# A Versatile Causal Discovery Framework to Allow Causally-Related Hidden Variables

**Xinshuai Dong**[*1]    **Biwei Huang** [*2]    **Ignavier Ng**[1]    **Xiangchen Song**[1]    **Yujia Zheng**[1]
**Songyao Jin**[4]    **Roberto Legaspi**[3]    **Peter Spirtes**[1]    **Kun Zhang**[1,4]
[1]Carnegie Mellon University
[2]University of California San Diego
[3]KDDI Research
[4]Mohamed bin Zayed University of Artificial Intelligence

## Abstract

Most existing causal discovery methods rely on the assumption of no latent confounders, limiting their applicability in solving real-life problems. In this paper, we introduce a novel, versatile framework for causal discovery that accommodates the presence of causally-related hidden variables almost everywhere in the causal network (for instance, they can be effects of observed variables), based on rank information of covariance matrix over observed variables. We start by investigating the efficacy of rank in comparison to conditional independence and, theoretically, establish necessary and sufficient conditions for the identifiability of certain latent structural patterns. Furthermore, we develop a Rank-based Latent Causal Discovery algorithm, RLCD, that can efficiently locate hidden variables, determine their cardinalities, and discover the entire causal structure over both measured and hidden ones. We also show that, under certain graphical conditions, RLCD correctly identifies the Markov Equivalence Class of the whole latent causal graph asymptotically. Experimental results on both synthetic and real-world personality data sets demonstrate the efficacy of the proposed approach in finite-sample cases. Our code will be publicly available.

## 1 Introduction and Related Work

Causal discovery aims at finding causal relationships from observational data and has received successful applications in many fields (Spirtes et al., 2000; 2010; Pearl, 2019). However, traditional methods, such as PC (Spirtes et al., 2000), GES (Chickering, 2002b), and LiNGAM (Shimizu et al., 2006b), generally assume that there are no latent confounders in the graph, which hardly holds in many real-world scenarios. Therefore, extensive efforts have been dedicated to addressing this issue for causal structure learning.

One line of research focuses on inferring the causal structure among the observed variables, despite the possible existence of latent confounders. Notable approaches include FCI and its variants (Spirtes et al., 2000; Pearl, 2000; Colombo et al., 2012; Akbari et al., 2021) that leverage conditional independence tests , and over-complete ICA-based techniques (Hoyer et al., 2008; Salehkaleybar et al., 2020) that further leverage non-Gaussianity.

Another line of thought focuses more on uncovering the causal structure among latent variables, by assuming observed variables are not directly adjacent. This includes Tetrad condition-based (Silva et al., 2006; Kummerfeld & Ramsey, 2016), high-order moments-based (Shimizu et al., 2009; Cai et al., 2019; Xie et al., 2020; Adams et al., 2021; Chen et al., 2022), matrix decomposition-based (Anandkumar et al., 2013), and mixture oracles-based (Kivva et al., 2021) approaches. Recently, Huang et al. (2022) propose an approach that makes use of rank constraints to identify general latent hierarchical structures, and yet observed variables can only be leaf nodes. Although Chandrasekaran et al. (2012) allow direct causal influences within observed variables and the existence of latent variables, it cannot recover the causal relationships among latent variables and has strong graphical constraints. For a more detailed discussion of related work, please refer to our Appx. C.1.

---

*Equal contribution.

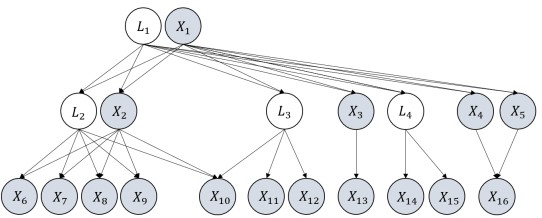

Figure 1: An example graph that we aim to handle, where latent variables are denoted by L and observed variables are denoted by X. The latent variables can act as a cause, effect, or mediator for both observed variables and other latent variables. See the appendix for more examples.

In this paper, we aim to handle a more general scenario for causal discovery with latent variables, where observed variables are allowed to be directly adjacent, and latent variables to be flexibly related to all the other variables. That is, hidden variables can serve as confounders, mediators, or effects of latent or observed variables, and even form a hierarchical structure (illustrated in Fig. 1). This setting is rather general and practically meaningful to deal with many real-world problems.

To address such a challenging problem, we are confronted with three fundamental questions: (i) What information and constraints can be discovered from the observed variables to reveal the underlying causal structure? (ii) How can we effectively and efficiently search for these constraints? (iii) What graphical conditions are needed to uniquely locate latent variables and ascertain the complete causal structure? Remarkably, these questions can be addressed by harnessing the power of rank deficiency constraints on the covariance of observed variables. By carefully identifying and utilizing rank properties in specific ways, we are able to determine the Markov equivalence class of the entire graph. Our contributions are mainly three-fold:

- We investigate the efficacy of rank in comparison to conditional independence in latent causal graph discovery, and theoretically introduce necessary and sufficient conditions for the identifiability of certain latent structural properties. For instance, the condition we proposed for nonadjacency generalizes the counterpart in Spirtes et al. (2000) to graphs with latent variables.
- We develop RLCD, an efficient three-phase causal discovery algorithm that is able to locate latent variables, determine their cardinalities, and identify the whole causal structure involving measured and latent variables, by properly leveraging rank properties. In the special case with no latent variables, it asymptotically returns the same graph as the PC algorithm (Spirtes et al., 2000) does.
- We provide a set of graphical conditions that are sufficient for RLCD to asymptotically identify the correct Markov Equivalence Class of the latent causal graph; notably, these graphical conditions are significantly weaker than those in previous works. Our empirical study on both synthetic and real-world datasets validates RLCD on finite samples.

## 2 PROBLEM SETTING

In this paper, we aim to identify the causal structure of a latent linear causal model defined as follows.

**Definition 1.** *(Latent Linear Causal Models) Suppose a directed acyclic graph* $\mathcal{G} := (\mathbf{V}_{\mathcal{G}}, \mathbf{E}_{\mathcal{G}})$, *where each variable* $V_i \in \mathbf{V}_{\mathcal{G}}$ *is generated following a linear causal structural model:*

$$\mathsf{V}_i = \sum\nolimits_{\mathsf{V}_j \in Pa_{\mathcal{G}}(\mathsf{V}_i)} a_{ij} \mathsf{V}_j + \varepsilon_{\mathsf{V}_i}, \tag{1}$$

*where* $\mathbf{V}_{\mathcal{G}} := \mathbf{L}_{\mathcal{G}} \cup \mathbf{X}_{\mathcal{G}}$ *contains a set of* $n + m$ *random variables, with* $m$ *latent variables* $\mathbf{L}_{\mathcal{G}} := \{\mathsf{L}_i\}_{i=1}^m$, *and* $n$ *observed variables* $\mathbf{X}_{\mathcal{G}} := \{\mathsf{X}_i\}_{i=1}^n$. $Pa_{\mathcal{G}}(V_i)$ *denotes the parent set of* $V_i$, $a_{ij}$ *the causal coefficient from* $V_j$ *to* $V_i$, *and* $\varepsilon_{\mathsf{V}_i}$ *represents the noise term.*

Our objective is to identify the underlying causal structure $\mathcal{G}$ over all the variables $\mathbf{L}_{\mathcal{G}} \cup \mathbf{X}_{\mathcal{G}}$ (detailed in Sec. 5) that are generated according to a latent linear causal model, given i.i.d. samples of observed variables $\mathbf{X}_{\mathcal{G}}$ only. To address this challenging problem, traditional wisdom often relies on strong graphical constraints (Pearl, 1988; Zhang, 2004; Huang et al., 2022; Maeda & Shimizu, 2020) (detailed in Appx. B.1 with illustrative graphs). In contrast, Definition 1 allows all the variables including observed and latent variables to be very flexibly related. We basically allow the presence of edges between any two variables such that a node V, no matter whether it is observed or not, can act as a cause, effect, or mediator for both observed and latent variables.

A summary of notations is in Tab. 1. The rest of the paper is organized as follows. In Sec. 3, we motivate the use of rank and propose conditions for nonadjacency and the existence of latent variables. In Sec. 4, we establish the minimal identifiable substructure of a linear latent graph, based on which we propose RLCD for latent variable causal discovery. In Sec. 5, we introduce the identifiability of RLCD. In Sec. 6, we validate our method using both synthetic and real-life data.

Table 1: Graphical notations used throughout this paper.

| Pa: Parents | $\mathbf{V}$: Variables | V: Variable | $\mathcal{G}$: The underlying graph |
|---|---|---|---|
| Ch: Children | $\mathbf{L}$: Latent variables | L: Latent variable | $\mathcal{G}'$: Output Graph |
| PCh: Pure children | $\mathbf{X}$: Observed variables | X: Observed variable | $\mathcal{S}$: Set of covers |
| Sib: Siblings | MDe: Measured descendants | PDe: Pure descendants | $\mathbb{S}$: Set of sets of covers |

## 3 Why Use Rank Information?

In this section, we first motivate the use of rank constraints for causal discovery in the presence of latent variables, and then establish some fundamental theories about what rank implies graphically.

### 3.1 Preliminaries about Treks and Rank

When there is no latent variable, a common approach for causal discovery is to use conditional independence (CI) relationships to identify d-separations in a graph; see, e.g., the PC algorithm (Spirtes et al., 2000). The following theorem illustrates this idea.

**Theorem 1** (Conditional Independence and D-separation (Pearl, 1988)). *Under the Markov and faithfulness assumption, for disjoint sets of variables* $\mathbf{A}$, $\mathbf{B}$ *and* $\mathbf{C}$, $\mathbf{C}$ *d-separates* $\mathbf{A}$ *and* $\mathbf{B}$ *in graph* $\mathcal{G}$, *iff* $\mathbf{A} \perp\!\!\!\perp \mathbf{B}|\mathbf{C}$ *holds for every distribution in the graphical model associated to* $\mathcal{G}$.

As for latent linear causal models, trek-separations (t-separations) provide more information than d-separations (for readers who are not very familiar with treks and t-separations, kindly refer to Appx. A.2 for examples). The definitions of treks and t-separation are given as follows, together with Theorem 2 showing the relations between t-separations and d-separations.

**Definition 2** (Treks (Sullivant et al., 2010)). *In* $\mathcal{G}$, *a trek from* X *to* Y *is an ordered pair of directed paths* $(P_1, P_2)$ *where* $P_1$ *has a sink* X, $P_2$ *has a sink* Y, *and both* $P_1$ *and* $P_2$ *have the same source* Z.

**Definition 3** (T-separation (Sullivant et al., 2010)). *Let* $\mathbf{A}$, $\mathbf{B}$, $\mathbf{C_A}$, *and* $\mathbf{C_B}$ *be four subsets of* $\mathbf{V}_\mathcal{G}$ *in graph* $\mathcal{G}$ *(not necessarilly disjoint).* $(\mathbf{C_A}, \mathbf{C_B})$ *t-separates* $\mathbf{A}$ *from* $\mathbf{B}$ *if for every trek* $(P_1, P_2)$ *from a vertex in* $\mathbf{A}$ *to a vertex in* $\mathbf{B}$, *either* $P_1$ *contains a vertex in* $\mathbf{C_A}$ *or* $P_2$ *contains a vertex in* $\mathbf{C_B}$.

**Theorem 2** (T- and D-sep (Di, 2009)). *For disjoint sets* $\mathbf{A}$, $\mathbf{B}$ *and* $\mathbf{C}$, $\mathbf{C}$ *d-separates* $\mathbf{A}$ *and* $\mathbf{B}$ *in graph* $\mathcal{G}$, *iff there is a partition* $\mathbf{C} = \mathbf{C_A} \cup \mathbf{C_B}$ *such that* $(\mathbf{C_A}, \mathbf{C_B})$ *t-separates* $\mathbf{A} \cup \mathbf{C}$ *from* $\mathbf{B} \cup \mathbf{C}$.

The theorem above reveals that all d-sep can be reformulated by t-sep, and thus, t-sep encompass d-sep information. Just as we use CI tests to find d-sep, t-sep can be identified by the rank of cross-covariance matrix over specific combinations of variables, which is formally stated as follows.

**Theorem 3** (Rank and T-separation (Sullivant et al., 2010)). *Given two sets of variables* $\mathbf{A}$ *and* $\mathbf{B}$ *from a linear model with graph* $\mathcal{G}$, *we have:*

$$rank(\Sigma_{\mathbf{A},\mathbf{B}}) \leq \min\{|\mathbf{C_A}| + |\mathbf{C_B}| : (\mathbf{C_A}, \mathbf{C_B}) \text{ t-separates } \mathbf{A} \text{ from } \mathbf{B} \text{ in } \mathcal{G}\}, \qquad (2)$$

*where* $\Sigma_{\mathbf{A},\mathbf{B}}$ *is the cross-covariance over* $\mathbf{A}$ *and* $\mathbf{B}$, *and equality generically holds.*

The left-hand side of Equation 2 is about properties of the observational distribution, while the right-hand side describes properties of the graph. Equality in Equation 2 generally holds, if there are no specific ways for the parameters of the underlying causal model to be coupled (e.g., accidental path cancellations). Recall that the classical faithfulness (Spirtes et al., 2000) is assumed so that we can infer d-separations from CI relations, here we formulate rank faithfulness in a similar spirit such that we can infer t-separations from rank deficiency constraints, as follows.

**Assumption 1** (Rank Faithfulness). *A probability distribution* $p$ *is rank faithful to* $\mathcal{G}$ *if the inequality in Equation 2 can be replaced by equality.*

This formulation of rank faithfulness differs from that of Spirtes (2013), though the spirit is shared The faithfulness assumption in causal discovery (Spirtes et al., 2000) is critical and prevalent. If faithfulness is violated (illustrated in Appx. B.2), even classical methods like PC cannot guarantee asymptotic correctness. On the other hand, with infinite data, the set of values of the SCM's free parameters s.t. rank faithfulness does not hold is of Lebesgue measure 0 (Spirtes, 2013).

Without latent variables, rank and CI are equally informative about the underlying DAG. Yet, in the presence of latent variables, t-separations inferred by rank offer more graphical information. Thus, we next show how rank constraint play a pivotal role in identifying latent causal structures.

### 3.2 Rank: An Informative Graphical Indicator for Latent Variables

In the presence of latent variables, CI is not enough: FCI and its variants make full use of CI but only recover a representation which is not informative enough about the latent confounders. Fortunately, leveraging rank information can naturally make causal discovery results more informative.

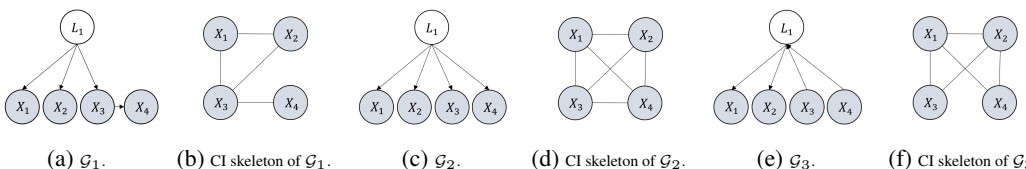

(a) $\mathcal{G}_1$.  (b) CI skeleton of $\mathcal{G}_1$.  (c) $\mathcal{G}_2$.  (d) CI skeleton of $\mathcal{G}_2$.  (e) $\mathcal{G}_3$.  (f) CI skeleton of $\mathcal{G}_3$.

Figure 2: Examples that illustrate the basic intuition for a latent to be identifiable.

An example highlighting the greater informativeness of rank compared to CI is as follows. Consider the graph $\mathcal{G}_1$ in Fig. 7, where $\{X_1, X_2\}$ and $\{X_3, X_4\}$ are d-separated by $L_1$, but we cannot infer that from a CI test (i.e., whether $\{X_1, X_2\} \perp\!\!\!\perp \{X_3, X_4\}|L_1$), as $L_1$ is not observed. In contrast, with rank information, we can infer that $rank(\Sigma_{\{X_1X_2\}, \{X_3X_4\}}) = 1$, which implies $\{X_1, X_2\}$ and $\{X_3, X_4\}$ are t-separated by one latent variable. The rationale behind is that the t-sep of $\mathbf{A}, \mathbf{B}$ by $(\mathbf{C_A}, \mathbf{C_B})$ can be deduced through rank (as in Theorem 3) without observing any element in $(\mathbf{C_A}, \mathbf{C_B})$.

With this intuition in mind, below we present three theorems that characterize the graphical implications of rank constraints, in scenarios where latent variables might exist: Theorem 4 gives conditions for observed variables to be nonadjacent, illustrated by Example 6; Theorem 5 gives conditions for the existence of latent variables, illustrated by Example 1; Theorem 6 implies how to utilize pure children as surrogates for calculating rank, illustrated with Example 7. All proofs are in Appendix.

**Theorem 4** (Condition for Nonadjacency). *Consider a latent linear causal model. Two observed variables* $X_1, X_2 \in \mathbf{X}_\mathcal{G}$ *are not adjacent, if there exist two sets* $\mathbf{A}, \mathbf{B} \subseteq \mathbf{X}_\mathcal{G} \setminus \{X_1, X_2\}$ *that are not necessarily disjoint, such that* $rank(\Sigma_{\mathbf{A} \cup \{X_1\}, \mathbf{B} \cup \{X_2\}}) = rank(\Sigma_{\mathbf{A}, \mathbf{B}})$ *and* $rank(\Sigma_{\mathbf{A} \cup \{X_1, X_2\}, \mathbf{B} \cup \{X_1, X_2\}}) = rank(\Sigma_{\mathbf{A}, \mathbf{B}}) + 2$.

**Remark 1.** *Theorem 4 presents a sufficient condition for determining nonadjacency between two observed variables. In the absence of latent variables, this condition becomes a both necessary and sufficient one. Note that* $\mathbf{A}$ *and* $\mathbf{B}$ *may overlap. SGS and PC (Spirtes et al., 2000) also introduced a necessary and sufficient condition for determining nonadjacency between two observed variables in the absence of latent variables: there exist a set of observed variables* $\mathbf{C} \subseteq \mathbf{X}_\mathcal{G}$, $X_1, X_2 \notin \mathbf{C}$, *such that* $X_1 \perp\!\!\!\perp X_2|\mathbf{C}$. *Interestingly, this condition can be expressed in the form of Theorem 4, with* $\mathbf{C} = \mathbf{A} = \mathbf{B}$; *thus Theorem 4 generalizes PC's condition to where latent variables may exist (detailed in Appx. A.8). Even in the presence of latent variables, with further graphical constraints the condition can also become a both necessary and sufficient one, as in Proposition 1.*

**Theorem 5** (Condition for Existence of Latent Variable). *Suppose a latent linear causal model with graph* $\mathcal{G}$ *and observed variables* $\mathbf{X}_\mathcal{G}$. *If there exist three disjoint sets of variables* $\mathbf{A}, \mathbf{B}, \mathbf{C} \subseteq \mathbf{X}_\mathcal{G}$, *such that (i)* $|\mathbf{B}| \geq |\mathbf{A}| \geq 2$, *(ii)* $\forall$ *distinct* $A_1, A_2 \in \mathbf{A}$, $A_1, A_2$ *are adjacent in the CI skeleton over* $\mathbf{X}_\mathcal{G}$ *with CI skeleton defined in Appx. A.3), (iii)* $\forall A \in \mathbf{A}, B \in \mathbf{B}$, $A, B$ *are adjacent in the CI skeleton over* $\mathbf{X}_\mathcal{G}$,*(iv)* $\mathbf{C} \subseteq \{X| \exists Y \in \mathbf{A} \cup \mathbf{B}$ *s.t.* $X, Y$ *are adjacent in the CI skeleton\} (i.e., all elements in* $\mathbf{C}$ *are neighbours of an element in* $\mathbf{A} \cup \mathbf{B}$ *in the CI skeleton), (v)* $rank(\Sigma_{\mathbf{A} \cup \mathbf{C}, \mathbf{B} \cup \mathbf{C}}) < |\mathbf{A}| + |\mathbf{C}|$, *then there must exist at least one latent variable in the treks between* $\mathbf{A}, \mathbf{B}$.

**Remark 2.** *Theorem 5 provides a sufficient condition for determining the existence of latent variables. The intuition is that, in the absence of latent variables, rank information should align with CI skeleton; if not, there must exist latent variable. Furthermore, we will show in Theorem 9 that, with further graphical constraints the condition in Theorem 5 becomes both necessary and sufficient.*

Moreover, it can be shown that observed children, or even descendants of latent variables can be used as surrogates to calculate rank as stated in Theorem 6 with the definition of *pure children* below.

**Definition 4** (Pure Children). *In* $\mathcal{G}$, *we denote the pure children set of a set of variables* $\mathbf{V}$, *as* $PCh_\mathcal{G}(\mathbf{V}) = \{C| Pa_\mathcal{G}(C) \neq \emptyset$ *and* $Pa_\mathcal{G}(C) \subseteq \mathbf{V}\}$.

**Theorem 6** (Pure Children as Surrogate for Rank Estimation). *Let* $\mathbf{C} \subseteq PCh_\mathcal{G}(\mathbf{A})$ *be a subset of pure children of* $\mathbf{A}$, *and* $\mathbf{B}$ *be a set of variables such that* $\mathbf{B} \cap De_\mathcal{G}(\mathbf{C}) = \emptyset$. *We have* $rank(\Sigma_{\mathbf{A}, \mathbf{B}}) \geq rank(\Sigma_{\mathbf{C}, \mathbf{B}})$. *Moreover, if* $rank(\Sigma_{\mathbf{A}, \mathbf{C}}) = |\mathbf{A}|$, *then* $rank(\Sigma_{\mathbf{A}, \mathbf{B}}) = rank(\Sigma_{\mathbf{C}, \mathbf{B}})$.

**Remark 3.** *Theorem 6 informs us that under certain conditions, we can estimate the rank of covariance involving latent variables by using their pure children as surrogates. Even when the pure children are not observed one can recursively examine the children's children until reaching observed descendants (defined in Appx. A.1). This enables us to deduce graphical information associated with latent variables through the use of observed ones as surrogates.*

## 4 DISCOVERING LATENT STRUCTURE THROUGH RANK CONSTRAINTS

In this section, we begin with the concept of *atomic cover* and explore its rank deficiency, and then develop an efficient algorithm based on rank-deficiency for latent causal discovery, as in Alg. 1.

### 4.1 ATOMIC COVER AND RANK DEFICIENCY

Below, we introduce *atomic covers* and their associated rank deficiency properties, which allows us to define the minimal identifiable substructure of a graph. We start with an example in Figure 2 that motivates the conditions for a latent variable to be identifiable.

**Example 1.** *Consider $\mathcal{G}_1$ in Fig. 2 (a). It can be shown that the latent variable $\mathsf{L}_1$ in $\mathcal{G}_1$ is not identifiable. We can easily find a graph $\mathcal{G}_1'$ with no latent variable, e.g., as in Fig. 10 (b), such that $\mathcal{G}_1'$ shares the same skeleton as Fig. 2 (a), but all the observational rank information entailed by $\mathcal{G}_1$ is the same as by $\mathcal{G}_1'$ - they are indistinguishable. However, if $\mathsf{X}_4$ becomes the children of $\mathsf{L}_1$, as in $\mathcal{G}_2$ given in Fig. 2 (c), the whole structure becomes identifiable. Specifically, the conditions in Theorem 5 holds when we take $\mathbf{A} = \{\mathsf{X}_1, \mathsf{X}_2\}$, $\mathbf{B} = \{\mathsf{X}_3, \mathsf{X}_4\}$, and $\mathbf{C} = \emptyset$, which informs the existence of latent variables. The same conditions also hold for Fig. 2 (e) and thus $\mathsf{L}_1$ in $\mathcal{G}_3$ is also identifiable, though $\mathsf{L}_1$ has only two children together with another two neighbors.*

The intuition is that, for a latent variable to be identifiable, it should have enough children and enough neighbors. We next formalize this intuition into the concept of *atomic cover* as the minimal identifiable unit in a graph. The formal definition of an atomic cover is given in Definition 5, where *effective cardinality* of a set of covers $\mathcal{V}$ is defined as $||\mathcal{V}|| = |(\cup_{\mathbf{V} \in \mathcal{V}} \mathbf{V})|$.

**Definition 5** (Atomic Cover and Atomic+$p$ Cover). *Let $\mathbf{V} \subseteq \mathbf{V}_{\mathcal{G}}$, where $l$ of the $|\mathbf{V}|$ variables are latent, and the remaining $|\mathbf{V}| - l$ are observed. $\mathbf{V}$ is an atomic+$p$ cover ($p \geq 0$), if $\mathbf{V}$ contains a single observed variable, or the following conditions hold:*

(i) *There exists a set of atomic+0 covers $\mathcal{C}$, s.t. $\cup_{\mathbf{C} \in \mathcal{C}} \mathbf{C} \subseteq PCh_{\mathcal{G}}(\mathbf{V})$ and $||\mathcal{C}|| \geq l + 1 + p$.*

(ii) *There exists a set of covers $\mathcal{N}$, s.t. $(\cup_{\mathbf{N} \in \mathcal{N}} \mathbf{N}) \cap (\cup_{\mathbf{C} \in \mathcal{C}} \mathbf{C}) = \emptyset$, every element in $\cup_{\mathbf{N} \in \mathcal{N}} \mathbf{N}$ is a neighbour of $\mathbf{V}$, $\mathcal{N}$ and $\mathcal{C}$ are d-separated by $\mathbf{V}$, and $||\mathcal{N}|| \geq l + 1$.*

(iii) *There does not exist a partition of $\mathbf{V} = \mathbf{V_1} \cup \mathbf{V_2}$, s.t., both $\mathbf{V_1}$ and $\mathbf{V_2}$ are atomic+0 covers.*

*For brevity atomic+0 cover is often referred to as atomic cover.*

We allow the definition to take different values of $p$ as we can benefit from more pure children for structure identification (as shown in Theorem 1). Yet, in this paper, we mainly focus on atomic covers (i.e., atomic+0 covers). We note that if $\mathbf{V}$ satisfies the definition of atomic+1 cover, it also satisfies the definition of atomic cover. In other words, a theorem that holds under atomic covers will also hold under atomic+1 covers. In the definition above, each observed variable is treated as an atomic cover, e.g., $\{\mathsf{X}_1\}$ in Figure 1. We define atomic cover as the minimal identifiable unit with the rationale that, when two or more latent variables share exactly the same set of neighbors (e.g., $\mathsf{L}_1$ and $\mathsf{L}_2$ in Fig. 6), they can never be distinguished from observational information. Hence, it is more convenient and unified to consider them together in an atomic cover. Examples of atomic covers can be found in Appx. B.3. Based on the definition of atomic covers, we define a *cluster* as the set of pure children of an atomic cover, and refer *k-cluster* to a cluster whose parents' cardinality is k. We further define an operator $Sep(\mathbf{X}) = \cup_{\mathsf{X} \in \mathbf{X}} \{\{\mathsf{X}\}\}$. We next show that every atomic cover possesses a useful rank deficiency property, which is formally stated in Theorem 7 (proof in Appx. A.11).

**Theorem 7** (Rank Deficiency of an Atomic Cover). *Let $\mathbf{V} = \mathbf{X} \cup \mathbf{L}$ be an atomic cover where $\mathbf{X}$ are observed, $\mathbf{L}$ are latent, and $|\mathbf{V}| = k$. Let $\mathcal{X} = Sep(\mathbf{X})$, $\mathcal{X}_{\mathcal{G}} = Sep(\mathbf{X}_{\mathcal{G}})$, and a set of atomic covers $\mathcal{C}$, satisfying (i) in Definition 5. Then $rank(\Sigma_{\mathcal{C} \cup \mathcal{X}, \mathcal{X} \cup \mathcal{X}_{\mathcal{G}} \setminus \mathcal{C} \setminus MDe_{\mathcal{G}}(\mathcal{C})}) = k$ and $k < \min(||\mathcal{C} \cup \mathcal{X}||, ||\mathcal{X} \cup \mathcal{X}_{\mathcal{G}} \setminus \mathcal{C} \setminus MDe_{\mathcal{G}}(\mathcal{C})||)$ ($MDe_{\mathcal{G}}$ denotes measured descendants).*

**Example 2** (Example for atomic cover and rank deficiency). *Consider an atomic cover in Fig. 4 (d): $\mathbf{V} = \{\mathsf{X}_2, \mathsf{L}_2\}$. $\mathbf{V}$ is an atomic cover, because $\mathbf{V}$ has at least 2 pure children and has additional 3 neighbors, satisfying the conditions in Definition 5. If we take $\mathcal{C} = \{\{\mathsf{X}_4\}, \{\mathsf{X}_5\}\}$, and $\mathcal{X} = \{\{\mathsf{X}_2\}\}$, we have $rank(\Sigma_{\mathcal{C} \cup \mathcal{X}, \mathcal{X} \cup \mathcal{X}_{\mathcal{G}} \setminus \mathcal{C} \setminus MDe_{\mathcal{G}}(\mathcal{C})}) = rank(\Sigma_{\{\mathsf{X}_4, \mathsf{X}_5, \mathsf{X}_2\}, \{\mathsf{X}_1, \mathsf{X}_2, \mathsf{X}_3, \mathsf{X}_6, \mathsf{X}_7, \mathsf{X}_8\}}) = 2 = |\mathbf{V}|$. Noted that both $||\mathcal{C} \cup \mathcal{X}||, ||\mathcal{X} \cup \mathcal{X}_{\mathcal{G}} \setminus \mathcal{C} \setminus MDe_{\mathcal{G}}(\mathcal{C})|| > 2$, so here the rank is deficient.*

Theorem 7 establishes the rank-deficiency property of an atomic cover. Furthermore, if we can build a unique connection between rank deficiency and atomic covers under certain conditions, then we can exploit rank deficiency to identify atomic covers in a graph. In the following, we present the graphical conditions to achieve identifiability and Theorem 8 delineates under what conditions the uniqueness of the rank-deficiency property can be ensured.

**Condition 1** (Basic Graphical Conditions for Identifiability). *Every latent variable belongs to at least one atomic cover in $\mathcal{G}$ and for each atomic cover that contains at least one latent variable, any of its children is not adjacent to any of its neighbours.*

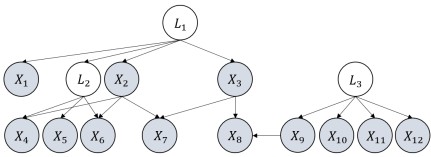

(a) The underlying graph $\mathcal{G}$, all the observed variables of which are taken as input to Phase 1.

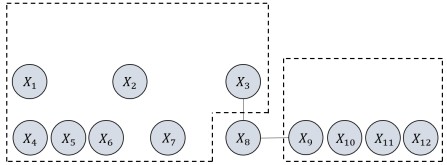

(b) Given the CI skeleton from Phase 1, variables are partitioned into two groups, shown in the two dashed areas.

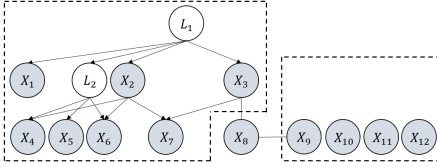

(c) Take variables from the first dashed area to Phase 2 and 3, use the result to update the CI skeleton.

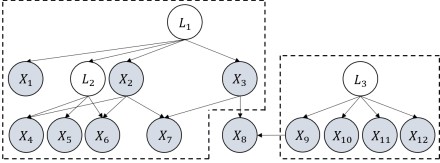

(d) Do the same thing as in (c) for the second dashed area and update all the remaining directions.

Figure 3: An illustrative example of the overall search procedure in Alg. 1.

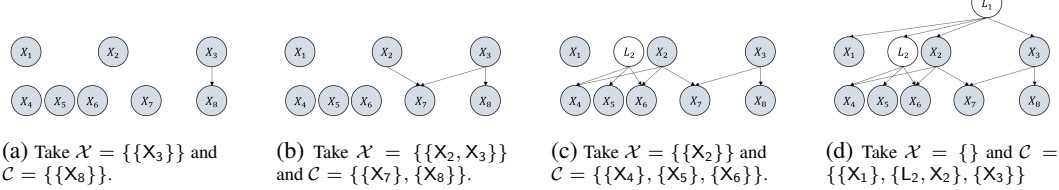

(a) Take $\mathcal{X} = \{\{X_3\}\}$ and $\mathcal{C} = \{\{X_8\}\}$.

(b) Take $\mathcal{X} = \{\{X_2, X_3\}\}$ and $\mathcal{C} = \{\{X_7\}, \{X_8\}\}$.

(c) Take $\mathcal{X} = \{\{X_2\}\}$ and $\mathcal{C} = \{\{X_4\}, \{X_5\}, \{X_6\}\}$.

(d) Take $\mathcal{X} = \{\}$ and $\mathcal{C} = \{\{X_1\}, \{L_2, X_2\}, \{X_3\}\}$

Figure 4: An illustrative example of the process of Phase 2 in Alg. 3.

**Theorem 8** (Uniqueness of Rank Deficiency). *Suppose $\mathcal{G}$ satisfies Condition 1. We further assume (i) all the atomic covers with cardinality $k' < k$ have been discovered and recorded, and (ii) there is no collider in $\mathcal{G}$. If there exists a set of observed variables $\mathbf{X}$ and a set of atomic covers $\mathcal{C}$ satisfying $\mathcal{X} = Sep(\mathbf{X})$, $\mathcal{C} \cap \mathcal{X} = \emptyset$, and $||\mathcal{C}|| + ||\mathcal{X}|| = k + 1$, such that (i) For all recorded $k'$ cluster $\mathcal{C}'$, $||\mathcal{C} \cap \mathcal{C}'|| \leq |Pa_{\mathcal{G}}(\mathcal{C}')|$, (ii) $rank(\Sigma_{\mathcal{C} \cup \mathcal{X}, \mathcal{X} \cup \mathcal{X}_{\mathcal{G}} \setminus \mathcal{C} \setminus MDe_{\mathcal{G}}(\mathcal{C})}) = k$, then there exists an atomic cover $\mathbf{V} = \mathbf{L} \cup \mathbf{X}$ in $\mathcal{G}$, with $|\mathbf{L}| = k - |\mathbf{X}|$, and $\cup_{\mathbf{C} \in \mathcal{C}} \mathbf{C} \subseteq PCh_{\mathcal{G}}(\mathbf{V})$.*

For a better understanding, we provide an illustrative example in Appx. A.12. Basically, Theorem 8 says that under certain conditions, we can build a unique connection between rank deficiency and atomic covers, and thus we can identify atomic covers in a graph by searching for combinations of $\mathcal{C}$ and $\mathcal{X}$ that induce rank deficient property. We further introduce Theorem 9, which is useful in that it provides necessary and sufficient conditions for the existence of latent variables, under Condition 1 and Theorem 1, which provides necessary and sufficient conditions for nonadjacency.

**Theorem 9** (Necessary and Sufficient Condition for Existence of Latent Variable). *If a graph satisfies Condition 1, then the "if" in Theorem 5 becomes "if and only if".*

**Proposition 1** (Necessary and Sufficient Condition for Nonadjacency). *In $\mathcal{G}$, if every latent variable belongs to at least one atomic+1 cover, then the "if" in Theorem 4 becomes "if and only if".*

Proposition 1 shows the further identifiability that can be established by assuming one additional pure child of atomic covers. As we mainly focus on atomic covers, the next question is, how to design a search procedure that strives to fulfill these conditions, in order to cash out Theorem 8 and Theorem 9, for identifying atomic covers, and consequently the whole latent causal structure.

## 4.2 METHOD - RANK-BASED LATENT CAUSAL DISCOVERY

In this section, we propose a computationally efficient and scalable search algorithm to identify the latent causal structure, referred to as *Rank-based Latent Causal Discovery (RLCD)*, that leverages the connection between graph structures and rank deficiency, as we discussed in previous sections. The search process mainly comprises three phases: (1) Phase 1: FindCISkeleton, (2) Phase 2: Find-CausalClusters, and (3) Phase 3: RefineCausalClusters, as outlined in Alg. 1.

Specifically, Phase 1 is to find the CI skeleton deduced by conditional independence tests over $\mathbf{X}_{\mathcal{G}}$, and Phase 2 and Phase 3 are designed such that the conditions in Theorem 8 are satisfied to the largest extent, in order to make use of the unique rank deficiency property to identify the latent structure. We initiate the process by finding the CI skeleton first, for the reason as follows. According to Theorem 9, latent variables exist iff conditions (i)-(v) in Theorem 5 are satisfied, while conditions

---

**Algorithm 1:** The overall procedure for Rank-based Latent Causal Discovery (RLCD).

---

**Input** : Samples from all $n$ observed variables $\mathbf{X}_{\mathcal{G}}$
**Output:** Markov equivalence class $\mathcal{G}'$

1 **def** *LatentVariableCausalDiscovery($\mathbf{X}_{\mathcal{G}}$):*
2      Phase 1: $\mathcal{G}' = $ FindCISkeleton($\mathbf{X}_{\mathcal{G}}$) (Algorithm 2);
3      **for** *Each $\mathcal{Q}$, a group of overlapping maximal cliques, in $\mathcal{G}'$* **do**
4          Set an empty graph $\mathcal{G}''$, $\mathbf{X}_{\mathcal{Q}} = \cup_{\mathbf{Q} \in \mathcal{Q}} \mathbf{Q}$, $\mathbf{N}_{\mathcal{Q}} = \{\mathsf{N} : \exists \mathsf{X} \in \mathbf{X}_{\mathcal{Q}} \text{ s.t. } \mathsf{N}, \mathsf{X} \text{ are adjacent in } \mathcal{G}'\}$;
5          Phase 2: $\mathcal{G}'' = $ FindCausalClusters($\mathcal{G}''$, $\mathbf{X}_{\mathcal{Q}} \cup \mathbf{N}_{\mathcal{Q}}$) (Algorithm 3);
6          Phase 3: $\mathcal{G}'' = $ RefineCausalClusters($\mathcal{G}''$, $\mathbf{X}_{\mathcal{Q}} \cup \mathbf{N}_{\mathcal{Q}}$) (Algorithm 4);
7          Transfer the estimated DAG $\mathcal{G}''$ to the Markov equivalence class and update $\mathcal{G}'$ by $\mathcal{G}''$;
8      Orient remaining causal directions that can be inferred from v structures;
9      **return** $\mathcal{G}'$

---

(i)-(iv) can be directed inferred from the CI skeleton. Therefore, there is no need to consider all variables in $\mathbf{X}_{\mathcal{G}}$ as inputs for Phases 2 and 3. Instead, we make use of the CI skeleton over $\mathbf{X}_{\mathcal{G}}$ and select some groups of observed variables as inputs into Phases 2 and 3, where variables in each group together have the potential to satisfy conditions (i)-(iv). This also benefits the computational efficiency as the Phase 2 and 3 for different groups can be done in parallel.

An example of the entire search process is illustrated in Fig. 3. After Phase 1, variables are partitioned into groups as shown in Fig. 3 (b). For each group (dashed area in Fig. 3 (b)), we conduct Phases 2 and 3, and have the final result shown in Fig. 3 (d). Details of each phase are given below.

### 4.3 PHASE 1: FINDING CI SKELETON

The objective of Phase 1 is to find the CI skeleton over $\mathbf{X}_{\mathcal{G}}$ by utilizing CI relations. To this end, we employ the first stage of the PC algorithm (Spirtes et al., 2000), with the difference that we replace all CI tests with rank tests, according to the following Lemma 10 (proof in Appx. A.16).

**Lemma 10** (D-separation by Rank Test). *Suppose a linear latent causal model with graph $\mathcal{G}$. For disjoint $\mathbf{A}, \mathbf{B}, \mathbf{C} \in \mathbf{X}_{\mathcal{G}}$, $\mathbf{C}$ d-separates $\mathbf{A}$ and $\mathbf{B}$ in graph $\mathcal{G}$, if and only if $rank(\Sigma_{\mathbf{A} \cup \mathbf{C}, \mathbf{B} \cup \mathbf{C}}) = |\mathbf{C}|$.*

We summarize the procedure of Phase 1 in Alg. 2 in the appendix. Although, asymptotically, using CI and rank information will provide the same d-separation result over observed variables, we use rank instead of CI in Phase 1 just for the purpose of having a unified causal discovery framework with rank constraints (as Phases 2 and 3 are also based on rank).

Given the CI skeleton $\mathcal{G}'$ (result from Phase 1), the next step is to find the substructures in $\mathcal{G}'$ that might contain latent variables. Specifically, Theorem 9 informs us that latent variable exists iff (i)-(v) in Theorem 5 holds, where (i)-(iv) can be directly inferred from the CI skeleton. Specifically, we consider all the maximal cliques $\mathbf{Q}$ in $\mathcal{G}'$ (a clique is a set of variables that are fully connected and a maximal clique is a clique that cannot be extended), s.t., $|\mathbf{Q}| \geq 3$. We then partition these cliques into groups such that two cliques $\mathbf{Q_1}, \mathbf{Q_2}$ are in the same group if $|\mathbf{Q_1} \cap \mathbf{Q_2}| \geq 2$. Finally, for each group of cliques $\mathcal{Q}$ (as in line 3 in Alg. 1), we combine them to form a set of variables $\mathbf{X}_{\mathcal{Q}} = \cup_{\mathbf{Q} \in \mathcal{Q}} \mathbf{Q}$. We further determine the neighbour set for each $\mathbf{X}_{\mathcal{Q}}$, as $\mathbf{N}_{\mathcal{Q}} = \{\mathsf{N} : \exists \mathsf{X} \in \mathbf{X}_{\mathcal{Q}} \text{ s.t. } \mathsf{N}, \mathsf{X} \text{ are adjacent in CI skeleton } \mathcal{G}'\}$. It can be shown that variables that satisfy (i)-(iv) will be in the same set with $\mathbf{X}_{\mathcal{Q}} \cup \mathbf{N}_{\mathcal{Q}}$ (examples and proof in Appx. B.4). Therefore our next step is to take each $\mathbf{X}_{\mathcal{Q}} \cup \mathbf{N}_{\mathcal{Q}}$ separately as input to our Phases 2 and 3, detailed as follows.

### 4.4 PHASE 2: FINDING CAUSAL CLUSTERS

In this section, we introduce the second phase of our algorithm, FindCausalClusters, summarized in Alg. 3 in the appendix and illustrated in Fig. 4. The objective here is to design an effective search procedure to find combinations of sets of covers $\mathcal{C}$ and $\mathcal{X}$ (as defined in Theorem 8), such that rank deficiency holds and all the conditions required in Theorem 8 are satisfied. To be specific, given Condition 1, Theorem 8 further requires that (i) when we are searching for $k$-clusters, all the $k'$-clusters, $k' < k$ have been found and recorded, and that (ii) there is no collider in $\mathcal{G}$. We next introduce the key designs for that end, accompanied by examples.

As for the requirement (i), we design our search procedure such that it starts with $k = 1$. If a k-cluster is found, we update the graph and reset $k$ to 1; otherwise, we increase $k$ by 1 (as in line 5 Alg 3). This ensures to a large extent that when searching for a $k$-cluster, all $k'$ clusters such that $k' < k$, can be found, in order to fulfill the requirement (i).

Regarding the requirement (ii), we need to ensure that all the rank deficiencies are from atomic covers, rather than from colliders. One could directly assume the absence of colliders in the underlying $\mathcal{G}$, but this would impose rather strong structural constraints and thus limit the applicability of a discovery method. Therefore, we add a collider check function *NoCollider* defined in Alg. 5, together with the designed search procedure to allow incorporating colliders timely such that they will not induce unexpected rank deficiency anymore. With these two designs, we only rely on a much weaker condition about colliders, as in Condition 2, under which it can be guaranteed that our search procedure will not be affected by the existence of colliders (proof in Appx. A.18).

**Condition 2** (Graphical condition on colliders for identifiability). *In a latent graph $\mathcal{G}$, if (i) there exists a set of variables $\mathbf{C}$ such that every variable in $\mathbf{C}$ is a collider of two atomic covers $\mathbf{V_1}, \mathbf{V_2}$, and denote by $\mathbf{A}$ the minimal set of variables that d-separates $\mathbf{V_1}$ from $\mathbf{V_2}$, (ii) there is a latent variable in $\mathbf{V_1}, \mathbf{V_2}, \mathbf{C}$ or $\mathbf{A}$, then we must have $|\mathbf{C}| + |\mathbf{A}| \geq |\mathbf{V_1}| + |\mathbf{V_2}|$.*

We summarize the whole process of Phase 2 in Alg. 3 and provide an illustration in Fig. 4, where the input variables are from the left dash area in Fig. 3 (b). For a better understanding, please refer to Appx. B.5 for a detailed description of key steps and illustrative examples.

### 4.5 PHASE 3: REFINING CAUSAL CLUSTERS

In Phase 2, we strive to fulfill all required conditions such that we can correctly identify causal clusters and related structures. However, there still exist some rare cases where our search cannot ensure the requirement (i) in Theorem 8. In this situation, Phase 2 might produce a big cluster in the resulting $\mathcal{G}'$ that should be split into smaller ones (see examples in Appx. B.6). Fortunately, the incorrect cluster will not do harm to the identification of other substructures in the graph, and thus we can employ Phase 3 to characterize and refine the incorrect ones, by making use of the following Theorem 11 (proof in Appx. A.17).

**Theorem 11** (Refining Clusters). *Denote by $\mathcal{G}'$ the output from FindCausalClusters and by $\mathcal{G}$ the true graph. For an atomic cover $\mathbf{V}$ in $\mathcal{G}'$, if $\mathbf{V}$ is not a correct cluster in $\mathcal{G}$ but consists of some smaller clusters, then $\mathbf{V}$ can be refined into correct ones by FindCausalClusters($\hat{\mathcal{G}}, \mathbf{X}$), where $\hat{\mathcal{G}}$ is got by deleting $\mathbf{V}$, all neighbors of $\mathbf{V}$ that are latent, and all relating edges of them, from $\mathcal{G}'$.*

To be specific, we search through all the atomic covers $\mathbf{V}$ in $\mathcal{G}'$, the output of Phase 2, and then perform FindCausalClusters($\hat{\mathcal{G}}, \mathbf{X}$), where $\hat{\mathcal{G}}$ is defined as in Theorem 11. With this procedure, we can make sure that all the found clusters in $\mathcal{G}'$ are correct as in $\mathcal{G}$. We summarized Phase 3 in Alg. 4 and provide illustrative examples in Appx. B.6.

## 5 IDENTIFIABILITY THEORY OF CAUSAL STRUCTURE

Here we show the identifiability of the proposed RLCD algorithm. Specifically, RLCD asymptotically produces the correct Markov equivalence class of the causal graph over both observed and latent variables under certain graphical conditions, up to the *minimal-graph operator* $\mathcal{O}_{\min}(\cdot)$ and *skeleton operator* $\mathcal{O}_s(\cdot)$. They are defined in Appx. A.4 following but differently with Huang et al. (2022). $\mathcal{O}_{\min}(\cdot)$ is to absorb redundant latent variables under certain conditions and $\mathcal{O}_s(\cdot)$ is to introduce edges involving latent variables if certain conditions hold, and we note that the observational rank information is invariant to these two graph operators (examples in Appx. B.12). We summarize the identifiability result in Theorem 12, along with Corollary 1 (proof in Appx. A.18).

**Theorem 12** (Identifiability of the Proposed Alg. 1). *Suppose $\mathcal{G}$ is a DAG associated with a Linear Latent Causal Model that satisfies Condition 1 and Condition 2. Algorithm 1 can asymptotically identify the Markov equivalence class of $\mathcal{O}_{min}(\mathcal{O}_s(\mathcal{G}))$.*

**Corollary 1.** *Assume linear causal models. Asymptotically, when no latent variable exists, Alg. 1's output is the same as that of PC; as another special case, when there is no edge between observed variables, the output of Alg. 1 is the same as that of Hier. rank (Huang et al., 2022).*

## 6 EXPERIMENTS

We validate our method using both synthetic and real-life data. In finite sample cases, we employ canonical correlations (Anderson, 1984) to estimate the rank (detailed in Appx. A.19).

**Synthetic Data** Specifically, we considered different types of latent graphs: (i) latent tree models (Appx. B.8), (ii) latent measurement models (Appx. B.9), and (iii) general latent models (Fig. 1 and Appx. B.10). The causal strength is uniformly from $[-10, 10]$, and the noise is either Gaussian or uniform (which is for RCD and GIN). We propose to use the following two metrics: **(1)** F1 score of

Table 2: F1 scores (mean (standard deviation)) of compared methods on different types of latent graphs.

| Algorithm | | Ours | Hier. rank | PC | FCI | GIN | RCD |
|---|---|---|---|---|---|---|---|
| | | **F1 score for skeleton among all variables $\mathbf{V}_{\mathcal{G}}$ (both $\mathbf{X}_{\mathcal{G}}$ and $\mathbf{L}_{\mathcal{G}}$)** | | | | | |
| *Latent+tree* | 2k | **0.84** (0.11) | 0.58 (0.01) | 0.36 (0.01) | 0.37 (0.01) | 0.37 (0.03) | 0.24 (0.04) |
| | 5k | **0.92** (0.05) | 0.60 (0.01) | 0.37 (0.00) | 0.37 (0.01) | 0.41 (0.03) | 0.33 (0.00) |
| | 10k | **0.98** (0.02) | 0.60 (0.01) | 0.37 (0.00) | 0.38 (0.02) | 0.41 (0.03) | 0.33 (0.01) |
| *Latent+measm* | 2k | **0.81** (0.12) | 0.52 (0.05) | 0.44 (0.01) | 0.38 (0.02) | 0.40 (0.02) | 0.26 (0.03) |
| | 5k | **0.88** (0.11) | 0.52 (0.05) | 0.49 (0.01) | 0.40 (0.01) | 0.46 (0.03) | 0.29 (0.01) |
| | 10k | **0.91** (0.09) | 0.53 (0.05) | 0.49 (0.00) | 0.40 (0.01) | 0.47 (0.05) | 0.34 (0.04) |
| *Latent general* | 2k | **0.66** (0.01) | 0.44 (0.02) | 0.31 (0.01) | 0.25 (0.02) | 0.30 (0.04) | 0.32 (0.03) |
| | 5k | **0.72** (0.03) | 0.45 (0.03) | 0.32 (0.01) | 0.28 (0.02) | 0.38 (0.04) | 0.34 (0.02) |
| | 10k | **0.80** (0.05) | 0.45 (0.04) | 0.32 (0.01) | 0.28 (0.02) | 0.35 (0.01) | 0.36 (0.01) |

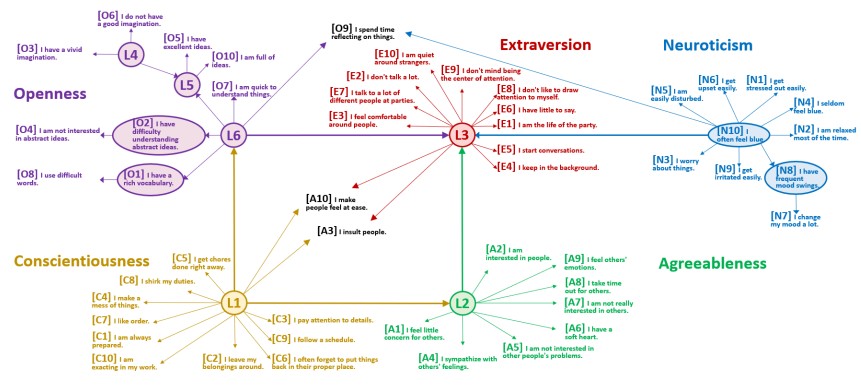

Figure 5: Identified latent graph on Big Five personality dataset.

skeleton among observed variables $\mathbf{X}_{\mathcal{G}}$, **(2)** F1 score of skeleton among all variables $\mathbf{V}_{\mathcal{G}}$. We consider combinations and permutations of latent variables during evaluation (detailed in Appx. B.15 together with definition of F1 score).

We compared with many competitive baselines, including **(i)** Hier. rank (Huang et al., 2022), **(ii)** PC (Spirtes et al., 2000), **(iii)** FCI (Spirtes et al., 2013), **(iv)** RCD (Maeda & Shimizu, 2020), and **(v)** GIN (Xie et al., 2020). The results are reported in Tab. 2 and Tab. 3, where we run experiments with different random seeds and sample sizes $2k$, $5k$, and $10k$. Our proposed RLCD gives the best results on all types of graphs, in terms of both metrics, with a clear margin. This result serves as strong empirical support for the identifiability of latent linear causal graphs by our proposed method.

**Real-World Data** To further verify our proposed method, we employed a real-world Big Five Personality dataset https://openpsychometrics.org/. It consists of 50 personality indicators and close to 20,000 data points. Each Big Five personality dimension, namely, Openness, Conscientiousness, Extraversion, Agreeableness, and Neuroticism (O-C-E-A-N), are measured with their own 10 indicators. Data is processed to have zero mean and unit variance. We employ the proposed method to determine the Markov equivalence class and employ GIN (Xie et al., 2020) to further decide other directions between latent variables (more details in Appendix B.17).

We analyzed the data using RLCD, producing a causal graph in Fig 5 that exhibits interesting psychological properties. First, most of the variables related to the same Big Five dimension are in the same cluster. Strikingly, our result reconciles two currently deemed distinct theories of personality: latent personality dimensions and network theory (Cramer et al., 2012; Wright, 2017). We see groups of closely connected items that are predictable from latent dimensions (L1, L2, L3), interactions among latents (L1→L6→L3, L1→L2→L3), and latents influencing the same indicators (L1, L3). We also observe plausible causal links between indicators (e.g., O2→O4 and O1→O8). We argue that our findings are consistent with pertinent personality literature, but more importantly, offer new, plausible explanations as to the nature of human personality (detailed analysis in Appx. B.18).

## 7 DISCUSSION AND CONCLUSION

We developed a versatile causal discovery approach that allows latent variables to be causally-related in a flexible way, by making use of rank information. We showed the proposed method can asymptotically identify the Markov equivalence class of the underlying graph under mild conditions. One limitation of our method is that it cannot directly handle cyclic graphs, and as future work, we will extend this line of thought to cyclic graphs and to nonlinear models.

ACKNOWLEDGEMENT

This material is based upon work supported by the AI Research Institutes Program funded by the National Science Foundation under AI Institute for Societal Decision Making (AI-SDM), Award No. 2229881. The project is also partially supported by the National Institutes of Health (NIH) under Contract R01HL159805, and grants from Apple Inc., KDDI Research Inc., Quris AI, and Infinite Brain Technology.

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

# Appendix

Organization of Appendices:

- Section A: Definitions, Examples, and Proofs
    - Section A.1: Subcovariance Matrix and Definition of Descendants.
    - Section A.2: Treks, T-separations, and Examples.
    - Section A.3: Definition of CI Skeleton
    - Section A.4: Definition of Rank-invariant Graph Operator.
    - Section A.5: Max-Flow-Min-Cut Lemma 13 for Treks
    - Section A.6: Example for Theorem 4.
    - Section A.7: Example for Theorem 6.
    - Section A.8: Proof of Theorem 4.
    - Section A.9: Proof of Theorem 5.
    - Section A.10: Proof of Theorem 6.
    - Section A.11: Proof of Theorem 7.
    - Section A.12: Example for Theorem 8.
    - Section A.13: Proof of Theorem 8.
    - Section A.14: Proof of Theorem 9.
    - Section A.16: Proof of Lemma 10.
    - Section A.17:Proof of Theorem 11.
    - Section A.18: Proof of Theorem 12.
    - Section A.19: Description of the rank test that we employ.
- Section B: Graphs, Illustrations of algorithms, and more information on datasets
    - Section B.1: Examples of Grpahs that Each Method Can Handle.
    - Section B.2: Example of Violation of faithfulness
    - Section B.3: Example of Atomic Cover.
    - Section B.4: Example for Phase 1
    - Section B.5: Detailed Description and Example for Phase 2.
    - Section B.6: Example for Phase 3.
    - Section B.7: Graph examples for model that has only observed variables.
    - Section B.8: Graph examples for latent tree model.
    - Section B.9: Graph examples for latent measurement model.
    - Section B.10: Graph examples for general latent model.
    - Section B.11: Example for considering colliders in Phase 1.
    - Section B.12: Examples for graph operators.
    - Section B.13: Examples for graphical relations between covers.
    - Section B.14: Discussions on checking colliders completely.
    - Section B.15: Evaluation metric details.
    - Section B.16: More details of experiments on synthetic data.
    - Section B.17: Detailed information of the Big Five personality dataset.
    - Section B.18: Detailed analysis of the result from the Big Five dataset.
- Section C: Related work
    - Section C.1: Related work.
- Section D: Additional Information (added during rebuttal)

Table 3: F1 scores (mean (standard deviation)) of compared methods on different types of latent graphs. F1 are calculated only for edges between observed variables $\mathbf{X}_\mathcal{G}$ in this graph. Hier.rank and GIN assume that observed variables are not directly adjacent so their performance is reported as -.

| Algorithm | | **F1 score for skeleton among $\mathbf{X}_\mathcal{G}$** | | | | | |
|---|---|---|---|---|---|---|---|
| | | Ours | Hier. rank | PC | FCI | GIN | RCD |
| *Latent+tree* | 2k | **0.79** (0.16) | - | 0.46 (0.02) | 0.47 (0.01) | - | 0.30 (0.03) |
| | 5k | **0.86** (0.10) | - | 0.44 (0.00) | 0.46 (0.02) | - | 0.38 (0.01) |
| | 10k | **0.97** (0.04) | - | 0.44 (0.00) | 0.46 (0.01) | - | 0.39 (0.02) |
| *Latent+measm* | 2k | **0.84** (0.11) | - | 0.50 (0.02) | 0.53 (0.02) | - | 0.30 (0.02) |
| | 5k | **0.93** (0.08) | - | 0.49 (0.01) | 0.51 (0.02) | - | 0.32 (0.02) |
| | 10k | **0.95** (0.05) | - | 0.48 (0.02) | 0.51 (0.02) | - | 0.42 (0.09) |
| *Latent general* | 2k | **0.68** (0.02) | - | 0.44 (0.01) | 0.39 (0.01) | - | 0.39 (0.06) |
| | 5k | **0.71** (0.03) | - | 0.45 (0.01) | 0.41 (0.04) | - | 0.44 (0.05) |
| | 10k | **0.78** (0.06) | - | 0.45 (0.01) | 0.41 (0.02) | - | 0.44 (0.01) |

# A    DEFINITIONS, EXAMPLES, AND PROOFS

## A.1    SUBCOVARIANCE MATRIX AND DEFINITION OF DESCENDANTS

$\Sigma_{\mathbf{A},\mathbf{B}}$ refers to subcovariance over $\mathbf{A}$ and $\mathbf{B}$. E.g., $\Sigma_{\{X_1,X_2\},\{X_3\}}$ is a $2 \times 1$ matrix whose entry at $(1,1)$ is $\mathrm{Cov}(X_1,X_3)$ and entry at $(2,1)$ is $\mathrm{Cov}(X_2,X_3)$. $\Sigma_{\mathcal{A},\mathcal{B}}$ refers to subcovariance over $\mathcal{A}$ and $\mathcal{B}$. Specifically,$\Sigma_{\mathcal{A},\mathcal{B}} = \Sigma_{\cup_{\mathbf{A}\in\mathcal{A}}\mathbf{A},\cup_{\mathbf{B}\in\mathcal{B}}\mathbf{B}}$. E.g., $\Sigma_{\{\{X_1\},\{X_2\}\},\{\{X_3\}\}}$ is also a $2 \times 1$ matrix whose entry at $(1,1)$ is $\mathrm{Cov}(X_1,X_3)$ and entry at $(2,1)$ is $\mathrm{Cov}(X_2,X_3)$.

As for the definition of descendants, a descendant is a node that can be reached by following one or more directed edges from a given node, and this definition excludes the node itself.

## A.2    TREKS, T-SEPARATIONS, AND EXAMPLES

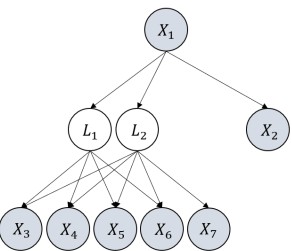

Figure 6: An example graph to show treks, t-separations, and rank of subcovariance matrix.

**Example 3** (Example of Treks). *In Figure 6, there are four treks between* $X_3$ *and* $X_4$: *(i)* $(P_1,P_2)=(X_3 \leftarrow L_1, L_1 \rightarrow X_4)$, *(ii)* $(P_1,P_2)=(X_3 \leftarrow L_2, L_2 \rightarrow X_4)$, *(iii)* $(P_1,P_2)=(X_3 \leftarrow L_1 \leftarrow X_1, X_1 \rightarrow L_2 \rightarrow X_4)$, *(iv)* $(P_1,P_2)=(X_3 \leftarrow L_2 \leftarrow X_1, X_1 \rightarrow L_1 \rightarrow X_4)$. *For adjacent variables such as* $X_1$ *and* $X_2$, *there must exist at least one trek* $(P_1,P_2)=(X_1, X_1 \rightarrow X_2)$ *between them.*

**Example 4** (Example of Trek-separations). *In Figure 6,* $X_3$ *and* $X_4$ *can be t-separated by* $(\{L_1,L_2\},\emptyset)$, *as for all the treks (i)-(iv) in Example 3, either* $P_1$ *contains a vertex in* $\{L_1,L_2\}$ *or* $P_2$ *contains a vertex in* $\emptyset$. *Similarly,* $X_3$ *and* $X_4$ *can also be t-separated by* $(\emptyset,\{L_1,L_2\})$. *However, the most simple way to t-separate* $X_3$ *and* $X_4$ *is by* $(\{X_3\},\emptyset)$ *or* $(\emptyset,\{X_4\})$.

**Example 5** (Example of calculating rank). *As shown in Example 4, the minimal way to t-separate* $X_3$ *and* $X_4$ *is by* $(\mathbf{C_A},\mathbf{C_B}) = (\{X_3\},\emptyset)$ *or* $(\emptyset,\{X_4\})$, *and thus* $\min|\mathbf{C_A}| + |\mathbf{C_B}| = 1$. *Therefore,* $rank(\Sigma_{\{X_3\},\{X_4\}}) = 1$. *Now suppose we want to calculate* $rank(\Sigma_{\{X_3,X_4,X_5\},\{X_1,X_6,X_7\}})$. *As the minimal way to t-separate* $\{X_3,X_4,X_5\}$ *and* $\{X_1,X_6,X_7\}$ *is by* $(\{L_1,L_2\},\emptyset)$ *(or* $(\emptyset,\{L_1,L_2\})$), *the rank is* $|\{L_1,L_2\}| + |\emptyset|=2$.

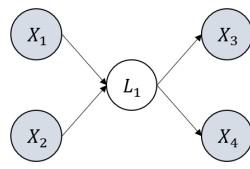

Figure 7: An illustrative example that highlights the motivation for using rank. When using CI, we cannot deduce that $\{X_1, X_2\}$ and $\{X_3, X_4\}$ are d-separated by $L_1$ as $L_1$ is latent, while by using rank we can.

### A.3 DEFINITION OF CI SKELETON

**Definition 6.** *A CI skeleton of $\mathbf{X}_{\mathcal{G}}$ is an undirected graph where the edge between $X_1$ and $X_2$ exists iff there does not exist a set of observed variables $\mathbf{C}$ such that $X_1, X_2 \notin \mathbf{C}$ and $X_1 \perp\!\!\!\perp X_2 | \mathbf{C}$.*

Examples of CI skeleton can be found in Example 2

### A.4 DEFINITION OF RANK-INVARIANT GRAPH OPERATOR

The definitions are as follows with examples in Appx. B.12.

**Definition 7** (Minimal-Graph Operator (Huang et al., 2022)). *Given two atomic covers $\mathbf{L}, \mathbf{P}$ in $\mathcal{G}$, we can merge $\mathbf{L}$ to $\mathbf{P}$ if the following conditions hold: (i) $\mathbf{L}$ is the pure children of $\mathbf{P}$, (ii) all elements of $\mathbf{L}$ and $\mathbf{P}$ are latent and $|\mathbf{L}| = |\mathbf{P}|$, and (iii) the pure children of $\mathbf{L}$ form a single atomic cover, or the siblings of $\mathbf{L}$ form a single atomic cover. We denote such an operator as minimal-graph operator $\mathcal{O}_{min}(\mathcal{G})$.*

**Definition 8** (Skeleton Operator). *Given an atomic covers $\mathbf{V}$ in a graph $\mathcal{G}$, we do the following two operations. (i) add directed edges such that $\forall V_1, V_2 \in \mathbf{V}$, $V_1$ and $V_2$ are adjacent and the whole graph is still acyclic. (ii) consider $\mathcal{S}$ as a set of atomic covers such that $\forall \mathbf{S} \in \mathcal{S}$, $\mathbf{S} \subseteq \mathbf{V}$. Let $\mathbf{C} = PCh_{\mathcal{G}}(\mathbf{V}) \backslash \cup_{\mathbf{S} \in \mathcal{S}} PCh_{\mathcal{G}}(\mathbf{S})$. We can draw edges from elements in $\mathbf{V}$ to elements in $\mathbf{C}$. We denote the combination of (i) and (ii) as the skeleton operator $\mathcal{O}_s(\mathcal{G})$.*

### A.5 MAX-FLOW-MIN-CUT LEMMA FOR TREKS

**Lemma 13** (Max-Flow-Min-Cut for Treks (Sullivant et al., 2010)). *The minimal $|\mathbf{C_A}| + |\mathbf{C_B}|$ s.t., $(\mathbf{C_A}|, |\mathbf{C_B}|)$ t-separates $\mathbf{A}$ from $\mathbf{B}$, equals to the maximum number of non-overlapping (no sided intersection) treks from $\mathbf{A}$ to $\mathbf{B}$.*

### A.6 EXAMPLE FOR THEOREM 4

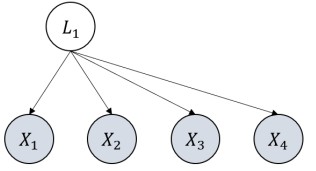

(a) Illustrative graph $\mathcal{G}_1$ for Theorem 4.

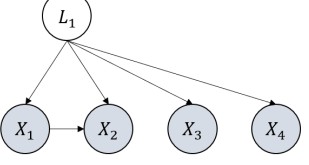

(b) Illustrative graph $\mathcal{G}_2$ for Theorem 4.

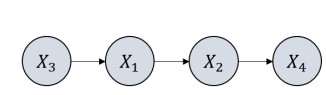

(c) Illustrative graph $\mathcal{G}_3$ for Theorem 4.

Figure 8: Illustrative figures for Theorem 4.

**Example 6.** *In Figure 8 (a), we can employ Theorem 4 to check whether $X_1$ and $X_2$ are adjacent. Specifically, let $\mathbf{A} = \{X_3\}$ and $\mathbf{B} = \{X_4\}$, then the condition in Theorem 4 is satisfied, i.e., $rank(\Sigma_{\mathbf{A} \cup \{X_1\}, \mathbf{B} \cup \{X_2\}}) = rank(\Sigma_{\mathbf{A}, \mathbf{B}})$ and $rank(\Sigma_{\mathbf{A} \cup \{X_1, X_2\}, \mathbf{B} \cup \{X_1, X_2\}}) = rank(\Sigma_{\mathbf{A}, \mathbf{B}}) + 2$. Therefore $X_1$ and $X_2$ are not adjacent.*

*When it comes to Figure 8 (b), the condition does not hold, and thus we cannot conclude that $X_1$ and $X_2$ are not adjacent.*

*Figure 8 (c) is an example to show that $rank(\Sigma_{\mathbf{A} \cup \{X_1, X_2\}, \mathbf{B} \cup \{X_1, X_2\}}) = rank(\Sigma_{\mathbf{A}, \mathbf{B}}) + 2$ in the condition is important in the theorem. We need this condition to ensure than treks between $\mathbf{A}$ and $\mathbf{B}$ do not rely on $X_1$ or $X_2$. Otherwise, as in Figure 8 (c), if we only check $rank(\Sigma_{\mathbf{A} \cup \{X_1\}, \mathbf{B} \cup \{X_2\}}) = rank(\Sigma_{\mathbf{A}, \mathbf{B}})$, we will found that it holds if we take $\mathbf{A} = \{X_3\}$ and $\mathbf{B} = \{X_4\}$. Therefore, we will mistakenly conclude that $X_1$ and $X_2$ are not adjacent.*

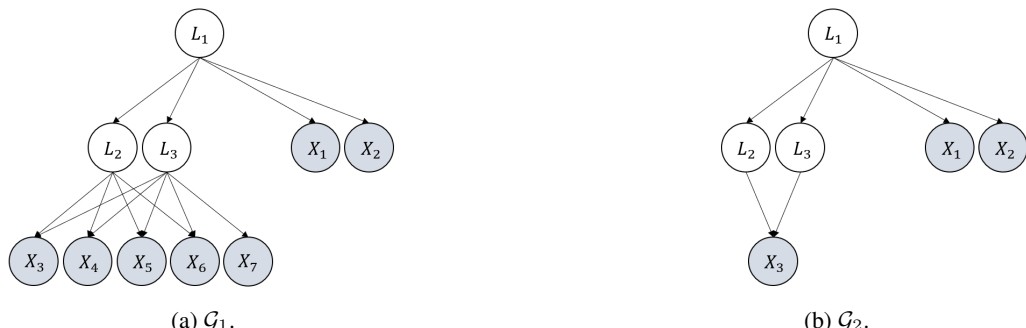

Figure 9: Illustrative figures for Theorem 6.

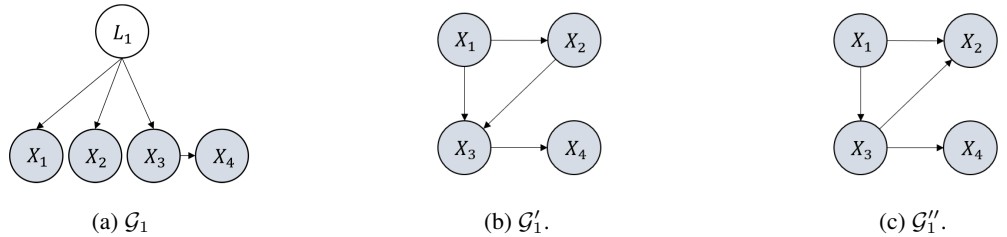

Figure 10: Examples to show that $\mathcal{G}_1$ cannot be identified as $\mathcal{G}_1$ $\mathcal{G}_1'$,$\mathcal{G}_1''$ have the same observational rank information.

### A.7 EXAMPLE FOR THEOREM 6

**Example 7.** *Take Figure 9 (a) as an example. Suppose we aim to calculate $rank(\Sigma_{\mathbf{A},\mathbf{B}})$, where $\mathbf{A} = \{\mathsf{L}_2, \mathsf{L}_3\}$ and $\mathbf{B} = \{\mathsf{X}_1, \mathsf{X}_2\}$. We can employ the pure children of $\mathbf{A}$, $\mathbf{C} = \{\mathsf{X}_3, ..., \mathsf{X}_7\}$ to do so. Specifically, we have $rank(\Sigma_{\mathbf{A},\mathbf{B}})=rank(\Sigma_{\mathbf{C},\mathbf{B}})= rank(\Sigma_{\{\mathsf{X}_3,...,\mathsf{X}_7\},\{\mathsf{X}_1,\mathsf{X}_2\}}) = 1$.*

*However, in Figure 9 (b), it is not the case. this is because the number of pure children of $\{\mathsf{L}_2, \mathsf{L}_3\}$ is not enough and thus $rank(\Sigma_{\mathbf{A},\mathbf{C}}) \neq |\mathbf{A}|$. In this case $\mathbf{C} = \{\mathsf{X}_3\}$ cannot work as a surrogate for calculating $rank(\Sigma_{\mathbf{A},\mathbf{B}})$.*

### A.8 PROOF OF THEOREM 4

*Proof.* Assume that the condition holds and suppose $\mathrm{rank}(\Sigma_{\mathbf{A},\mathbf{B}}) = t$, which means that there exist $t$ non-overlapping treks between $\mathbf{A}$ and $\mathbf{B}$. By $\mathrm{rank}(\Sigma_{\mathbf{A}\cup\{\mathsf{X}_1,\mathsf{X}_2\},\mathbf{B}\cup\{\mathsf{X}_1,\mathsf{X}_2\}}) = \mathrm{rank}(\Sigma_{\mathbf{A},\mathbf{B}}) + 2$, we further have that these $t$ treks do not necessarily travel across $\mathsf{X}_1$ or $\mathsf{X}_2$. As such, if $\mathsf{X}_1$ and $\mathsf{X}_2$ are adjacent, then we must have $\mathrm{rank}(\Sigma_{\mathbf{A}\cup\{\mathsf{X}_1\},\mathbf{B}\cup\{\mathsf{X}_2\}}) = t+1 \neq \mathrm{rank}(\Sigma_{\mathbf{A},\mathbf{B}})$, which contradicts with the condition. Therefore $\mathsf{X}_1$ and $\mathsf{X}_2$ cannot be adjacent. $\qquad\square$

Here we show that the condition generalizes PC's condition. SGS and PC (Spirtes et al., 2000) proposed: there exist a set of observed variables $\mathbf{C} \subseteq \mathbf{X}_{\mathcal{G}}$, $\mathsf{X}_1, \mathsf{X}_2 \notin \mathbf{C}$, such that $\mathsf{X}_1 \perp\!\!\!\perp \mathsf{X}_2 | \mathbf{C}$. This condition can be expressed in the form of Theorem 4, with $\mathbf{C} = \mathbf{A} = \mathbf{B}$, because we have $\mathsf{X}_1 \perp\!\!\!\perp \mathsf{X}_2 | \mathbf{C}$ iff $\mathrm{rank}(\Sigma_{\mathbf{A}\cup\{\mathsf{X}_1\},\mathbf{B}\cup\{\mathsf{X}_2\}}) = \mathrm{rank}(\Sigma_{\mathbf{A},\mathbf{B}})$. Plus, $\mathrm{rank}(\Sigma_{\mathbf{A}\cup\{\mathsf{X}_1,\mathsf{X}_2\},\mathbf{B}\cup\{\mathsf{X}_1,\mathsf{X}_2\}}) = \mathrm{rank}(\Sigma_{\mathbf{A},\mathbf{B}}) + 2$ is always true when $\mathbf{A} = \mathbf{B}$.

### A.9 PROOF OF THEOREM 5

We first introduce Lemma 14 as follows to show that when there is no latent variable, the rank information should be aligned with what CI skeleton provides.

**Lemma 14.** *When there is no latent variable in the graph, rank and CI are equally informative about the underlying structure, i.e., the rank-equivalence class and the Markov equivalence class are the same when there is no latent variable.*

*Proof.* (i) As all d-sep can be stated by rank according to Lemma 10, using rank information is able to arrive at the markov equivalence class. (ii) Every element in the markov equivalence class are

distributionally equivalent in terms of second order statistics. Therefore, using information from the rank of the covariance matrix cannot differentiate elements in the markove equivalence class. Taking (i) and (ii) together, we have that the rank-equivalence class and the Markov equivalence class are the same when there is no latent variable. $\qquad\square$

Bellow is the proof of Theorem 5.

*Proof.* First we assume that there is no latent variable and we have that the CI skeleton among observed variables is also the skeleton of the true underlying graph $\mathcal{G}$. By (ii) and (iii), we have (ii') $\forall$ distinct $A_1, A_2 \in \mathbf{A}$, $A_1, A_2$ are adjacent in $\mathcal{G}$, (iii') $\forall A \in \mathbf{A}, B \in \mathbf{B}$, $A, B$ are adjacent in $\mathcal{G}$. By (ii') and (iii'), it must hold that (iv') $\mathrm{rank}(\Sigma_{\mathbf{A}\cup\mathbf{C},\mathbf{B}\cup\mathbf{C}}) = |\mathbf{A}| + |\mathbf{C}|$. However, (iv') contradicts with (iv). Therefore, there must exist at least one latent variable. $\qquad\square$

### A.10 Proof of Theorem 6

*Proof.* By Theorem 3 and Lemma 13, we have $\mathrm{rank}(\Sigma_{\mathbf{A},\mathbf{B}})$ equals the maximum number of non-overlapping trek paths from $\mathbf{A}$ to $\mathbf{B}$. As $\mathbf{C}$ are pure children of $\mathbf{A}$ and no element in $\mathbf{B}$ are descendants of $\mathbf{C}$, every trek from $\mathbf{C}$ to $\mathbf{B}$ must travel across $\mathbf{A}$. Therefore, $\mathrm{rank}(\Sigma_{\mathbf{A},\mathbf{B}}) \geq \mathrm{rank}(\Sigma_{\mathbf{C},\mathbf{B}})$. If we further have $\mathrm{rank}(\Sigma_{\mathbf{A},\mathbf{C}}) = |\mathbf{A}|$, then the maximum number of non-overlapping trek paths from $\mathbf{A}$ to $\mathbf{B}$ equals the maximum number of non-overlapping trek paths from $\mathbf{C}$ to $\mathbf{B}$, and thus $\mathrm{rank}(\Sigma_{\mathbf{A},\mathbf{B}}) = \mathrm{rank}(\Sigma_{\mathbf{C},\mathbf{B}})$. $\qquad\square$

### A.11 Proof of Theorem 7

*Proof.* As none of elements in $\mathcal{X}_{\mathcal{G}} \setminus \mathcal{C}$ are descendants of $\mathcal{C}$, by Theorem 3, the minimal way to block every trek path between $\mathcal{C} \cup \mathcal{X}$ and $(\mathcal{X}_{\mathcal{G}} \setminus \mathcal{C}) \cup \mathcal{X}$ is by blocking all the elements of the atomic cover $\mathbf{V}$ and all the elements in $\mathbf{X}$. As $\mathbf{X}$ is a subset of $\mathbf{V}$, we have $\mathrm{rank}(\Sigma_{\mathcal{C}\cup\mathcal{X},(\mathcal{X}_{\mathcal{G}}\setminus\mathcal{C})\cup\mathcal{X}}) = |\mathbf{V}| = k$. $\qquad\square$

### A.12 Example for Theorem 8

**Example 8** (Example for the uniqueness of rank deficiency in Theorem 8). *In Figure 1, if the current $k = 2$, and we assume that all $k = 1$ clusters are found, and no v-structure exists, the rank deficiency would uniquely map to a $k = 2$ cluster. E.g., if we take $\mathcal{C} = \{\{X_6\}, \{X_7\}\}$ and $\mathcal{X} = \{\{X_2\}\}$, we have $\mathrm{rank}(\Sigma_{\mathcal{C}\cup\mathcal{X},\mathcal{X}\cup\mathcal{X}_{\mathcal{G}}\setminus\mathcal{C}\setminus\mathrm{MDe}_{\mathcal{G}}(\mathcal{C})}) = \mathrm{rank}(\Sigma_{\{X_6,X_7,X_2\},\{X_1,...,X_5,X_8,...,X_{16}\}}) = 2$. In this case, this rank deficiency uniquely relates to a cover $\mathbf{V} = \mathbf{X} \cup \mathbf{L}$, where $\mathbf{X} = \cup_{\mathbf{X}'\in\mathcal{X}}\mathbf{X}' = \{X_2\}$, $\mathbf{L}$ is latent variable to be added with $|\mathbf{L}| = k - |\mathbf{X}| = 1$, and $\mathcal{C} = \{\{X_6\}, \{X_7\}\}$ are the pure children of $\mathbf{V}$.*

*In contrast, if we are searching for $k = 2$ and a 1-cluster $L_4 \to \{X_{14}, X_5\}$ has not been identified, then the condition for Theorem 8 is not satisfied and thus the uniqueness of rank deficiency does not hold: e.g., by taking $\mathcal{C} = \{X_{13}, X_{14}, X_{15}\}$ and $\mathcal{X} = \{\}$, we have $\mathrm{rank}(\Sigma_{\{X_{13},X_{14},X_{15}\},\{X_1,...,X_{12},X_{16}\}}) = 2$, which is deficient, and yet $\{X_{13}, X_{14}, X_{15}\}$ are not from a $k = 2$ cluster. This is because this rank deficiency is not from a $k = 2$ cluster, rather, it is from the 1-cluster $\{X_{14}, X_{15}\}$ with parent $L_4$ that has not been found yet.*

### A.13 Proof of Theorem 8

*Proof.* We first show that (a) if $||\mathcal{X}|| = t = 0$ and, elements from $\mathcal{C}$ are pure children of two or more atomic covers, then we must have $\mathrm{rank}(\Sigma_{\mathcal{C}\cup\mathcal{X},\mathcal{X}\cup\mathcal{X}_{\mathcal{G}}\setminus\mathcal{C}\setminus\mathrm{MDe}_{\mathcal{G}}(\mathcal{C})}) = k + 1$.

Proof of (a). As there is no collider between atomic covers, we have that all the elements from $\mathcal{C}$ are pure children of atomic covers. Suppose $\mathcal{C}$ can be partitioned into $\mathcal{C}_1, ..., \mathcal{C}_N$, where each $\mathcal{C}_i$ are the pure children of a distinct atomic cover in $\mathcal{G}$. If $\mathcal{C}_i$ are the pure children of an atomic cover with cardinality $k' < k$, then we have $||\mathcal{C}_i|| \leq k'$. If $\mathcal{C}_i$ are the pure children of an atomic cover with cardinality $k' \geq k \geq ||\mathcal{C}_i||$ ($k \geq ||\mathcal{C}_i||$ because if $k < ||\mathcal{C}_i||$ then all elements of $\mathcal{C}$ are from the same cluster), so we also have $||\mathcal{C}_i|| \leq k'$. Therefore, by Lemma 13 and the fact that each atomic cover has $k + 1 - t$ pure children and $k + 1$ additional neighbors, we have $\mathrm{rank}(\Sigma_{\mathcal{C}\cup\mathcal{X},\mathcal{X}\cup\mathcal{X}_{\mathcal{G}}\setminus\mathcal{C}\setminus\mathrm{MDe}_{\mathcal{G}}(\mathcal{C})}) = \mathrm{rank}(\Sigma_{\mathcal{C},\mathcal{X}_{\mathcal{G}}\setminus\mathcal{C}\setminus\mathrm{MDe}_{\mathcal{G}}(\mathcal{C})}) = \sum_1^N ||\mathcal{C}_i|| = k + 1$. Therefore, when $||\mathcal{X}|| = t = 0$, the rank deficiency property does not hold when elements of $\mathcal{C}$ are from different clusters.

(b) When $||\mathcal{X}|| = t \neq 0$, we consider a new graph $\mathcal{G}''$, where all variables from $||\mathcal{X}||$ are removed (as well as related edges). Assume elements of $\mathcal{C}$ are from different clusters and elements of $\mathcal{X}$ are from the same atomic cover. Thus by (a), we have that the maximum number of non-overlapping treks in $\mathcal{G}''$ between $\mathcal{C}$ and $\mathcal{X}_{\mathcal{G}''} \setminus \mathcal{C} \setminus \mathrm{MDe}_{\mathcal{G}''}(\mathcal{C})$ is $k + 1 - t$. Then we add $\mathcal{X}$ with relating edges back to the graph and thus we will have $||\mathcal{X}|| = t$ additional non-overlapping treks between $\mathcal{C} \cup \mathcal{X}$ and $\mathcal{X} \cup \mathcal{X}_{\mathcal{G}} \setminus \mathcal{C} \setminus \mathrm{MDe}_{\mathcal{G}}(\mathcal{C})$. Therefore, we also have $\mathrm{rank}(\Sigma_{\mathcal{C} \cup \mathcal{X}, \mathcal{X} \cup \mathcal{X}_{\mathcal{G}} \setminus \mathcal{C} \setminus \mathrm{MDe}_{\mathcal{G}}(\mathcal{C})}) = k + 1$, when $||\mathcal{X}|| = t \neq 0$ and elements of $\mathcal{C}$ are from different clusters. Similarly, if elements of $\mathcal{C}$ are from the same cluster but not all elements of $\mathcal{X}$ are the parents of that cluster, we also have $\mathrm{rank}(\Sigma_{\mathcal{C} \cup \mathcal{X}, \mathcal{X} \cup \mathcal{X}_{\mathcal{G}} \setminus \mathcal{C} \setminus \mathrm{MDe}_{\mathcal{G}}(\mathcal{C})}) = k + 1$.

Taking (a) and (b) together, we have that rank deficiency holds only if all elements of $\mathcal{C}$ are from the same cluster and all elements of $\mathcal{X}$ are the parents of that cluster. $\qquad\square$

### A.14 PROOF OF THEOREM 9

In the proof of Theorem 5, we have already shown the 'if' direction. Now we are going to show the sketch of the proof for the 'only if' direction.

*Proof.* Suppose there is a latent variable in $\mathcal{G}$. According to Condition 1, it must belong to an atomic cover, say $\mathbf{V}$ and we suppose that $\mathbf{V}$ contains $n$ latent variables in total, rest of which are observed $\mathbf{X}$. According to the definition of atomic cover, $\mathbf{V}$ has at least $n + 1$ pure children and $n + 1$ neighbours that are distinct with the $n + 1$ pure children. Assume that all of them are observed. Then we can simply take $\mathbf{A}$ as the pure children, $\mathbf{B}$ as the neighbours, $\mathbf{C}$ as $\mathbf{X}$, and thus conditions (i)-(v) will be all satisfied. If some of the pure children or neighbors of $\mathbf{V}$ are latent, we can simply use their pure children instead (if the pure children are still latent, use the pure children of the pure children and finally we will find enough observed pure children/descendants, as latent variables cannot be leaf nodes). Thus the conditions (i)-(v) can also be satisfied. Therefore, if there is at least a latent variable, then there must exist disjoint $\mathbf{A}, \mathbf{B}$, and $\mathbf{C}$, such that (i)-(v) hold. $\qquad\square$

### A.15 PROOF OF THEOREM 1

In the proof of Theorem 1, we have already shown the 'if' direction. Now we are going to show the sketch of the proof for the 'only if' direction.

*Proof.* $\qquad\square$

### A.16 PROOF OF LEMMA 10

*Proof.* By Theorem 2 we have that for disjoint $\mathbf{A}, \mathbf{B}$ and $\mathbf{C}$, $\mathbf{C}$ d-separates $\mathbf{A}$ from $\mathbf{B}$, iff there is a partition $\mathbf{C} = \mathbf{C_A} \cup \mathbf{C_B}$ such that $(\mathbf{C_A}, \mathbf{C_B})$ t-separates $\mathbf{A} \cup \mathbf{C}$ from $\mathbf{B} \cup \mathbf{C}$. By Theorem 3, we have that $\mathrm{rank}(\Sigma_{\mathbf{A} \cup \mathbf{C}, \mathbf{B} \cup \mathbf{C}}) \leq |\mathbf{C}|$. Plus, by the definition of treks, $\mathrm{rank}(\Sigma_{\mathbf{A} \cup \mathbf{C}, \mathbf{B} \cup \mathbf{C}}) \geq |\mathbf{C}|$. Therefore, $\mathbf{C}$ d-separates $\mathbf{A}$ from $\mathbf{B}$, iff $\mathrm{rank}(\Sigma_{\mathbf{A} \cup \mathbf{C}, \mathbf{B} \cup \mathbf{C}}) = |\mathbf{C}|$.

$\qquad\square$

### A.17 PROOF OF THEOREM 11

*Proof.* The sketch of the proof is as follows.

We first show that a fake cover will not influence all other found structures except itself and its neighbors in the result $\mathcal{G}'$. By Lemma 11 in (Huang et al., 2022), we have that a fake cover with observed descendants $\mathbf{X}$ in $\mathcal{G}'$ implies a bond set in $\mathcal{G}$ (whose definition can be found in Huang et al. (2022)), and there is a partition of the rest of the observed variables $\mathbf{X}_{\mathcal{G}} \setminus \mathbf{X}$ into two groups $\mathbf{A}$ and $\mathbf{B}$ such that $\mathbf{A}$ and $\mathbf{B}$ are d-separated by the bond set. Suppose this faker cover is $\mathbf{V}$ that corresponds to a set of $n$ latent covers $\{\mathbf{L_1}, ..., \mathbf{L_n}\}$ in $\mathcal{G}$. Then $\{\mathbf{L_1}, ..., \mathbf{L_n}\}$ d-separates $\mathbf{A}$ and $\mathbf{B}$, and thus during the search we will generate two dummy covers that interact with $\mathbf{A}$ and $\mathbf{B}$ respectively, during which the rank information will not be mistaken. Then we show that in Phase 3, the fake cover can be corrected. When we refine the fake cover $\mathbf{V}$, we will delete it together with its neighbours and thus the two dummy covers will be deleted. Suppose now we are searching for $k$ clusters, as this

time all the remaining $k'$-clusters s.t. $k' < k$ have been found, the FindCausalCluster function will not generate fake cluster anymore. Therefore, the output would be corrected.

$\square$

### A.18 Proof of Theorem 12

First, we show an extension of Theorem 6 to atomic covers.

**Lemma 15** (Pure Children as Surrogate for Atomic Covers). *Let* $\mathbf{A} = \mathbf{L} \cup \mathbf{X}$ *be an atomic cover,* $\mathcal{C} \subseteq PCh_{\mathcal{G}}(\mathbf{A})$ *be a subset of pure children of* $\mathbf{A}$, *and* $\mathbf{B}_1, \mathbf{B}_2$ *be two sets of variables such that for all* $\mathsf{B} \in \mathbf{B}_1 \cup \mathbf{B}_2$, $\mathsf{B} \notin De_{\mathcal{G}}(\mathcal{C})$. *We have* $rank(\Sigma_{\{\mathbf{A}\} \cup \{\mathbf{B}_1\}, \mathbf{B}_2}) = rank(\Sigma_{\mathcal{C} \cup \{\mathbf{X}\} \cup \{\mathbf{B}_1\}, \mathbf{B}_2})$, *if* $||\mathcal{C}|| + |\mathbf{X}| \geq |\mathbf{A}|$.

*Proof.* By Lemma 13, $rank(\Sigma_{\{\mathbf{A}\} \cup \{\mathbf{B}_1\}, \mathbf{B}_2})$ is the max number of non-overlapping treks between $\{\mathbf{A}\} \cup \{\mathbf{B}_1\}$ and $\mathbf{B}_2$. If $||\mathcal{C}|| + |\mathbf{X}| \geq |\mathbf{A}|$ holds, all the treks starting from $\{\mathbf{A}\} \cup \{\mathbf{B}_1\}$ can be extended to treks that start from $\mathcal{C} \cup \{\mathbf{X}\} \cup \{\mathbf{B}_1\}$, and the max number of non-overlapping treks is the same, which means $rank(\Sigma_{\{\mathbf{A}\} \cup \{\mathbf{B}_1\}, \mathbf{B}_2}) = rank(\Sigma_{\mathcal{C} \cup \{\mathbf{X}\} \cup \{\mathbf{B}_1\}, \mathbf{B}_2})$. $\square$

This lemma informs us that if we correctly found a cover $\mathbf{A} = \mathbf{L} \cup \mathbf{X}$ by our rules in Algorithm 2, we can calculate the rank relating to $\mathbf{A}$ by using its pure children $\mathcal{C}$ together with part of the observation of $\mathbf{A}$, i.e., $\mathbf{X}$ as surrogates, even though part of $\mathbf{A}$, i.e., $\mathbf{L}$, cannot be observed. Note that $||\mathcal{C}|| + |\mathbf{X}| \geq |\mathbf{A}|$ always holds as it is required when we are searching for clusters in Algorithm 2.

Next, we show that when we are searching for combinations of $\mathcal{C}$ and $\mathcal{X}$ in Algorithm 2, by leveraging the checking function NoCollider defined in Algorithm 5, the correctness of Algorithm 2 will not be influenced even though there might exist colliders in $\mathcal{C}$.

**Lemma 16** (Colliders in $\mathcal{C}$ do not harm). *Suppose there exist some collider structures in graph* $\mathcal{G}$, *e.g., there exist two atomic covers* $\mathbf{V}_1$ *and* $\mathbf{V}_2$, *with* $\mathbf{A}$ *the minimal set of variables that d-separates* $\mathbf{V_1}$ *from* $\mathbf{V_2}$, *and* $\mathbf{C}$ *as a collider of* $\mathbf{V_1}, \mathbf{V_2}$. *The correctness of Algorithm 2 will not be influenced by the existence of colliders in* $\mathcal{C}$.

*Proof.* In Algorithm 2, we check whether different combinations of $\mathcal{C}$, $\mathcal{X}$, and $\mathcal{N}$ induce rank deficiency. Suppose we take $\mathcal{C} = \{\{\mathbf{V'}_1\}, \{\mathbf{V'}_2\}, \{\mathbf{C'}\}\}$, where $\mathbf{V'}_1 \subseteq \mathbf{V}_1$, $\mathbf{V'}_2 \subseteq \mathbf{V}_2$, and $\mathbf{C'} \subseteq \mathbf{C}$ and let $\mathbf{R}$ be $\mathbf{V}_1 \cup \mathbf{V}_2 \backslash \mathbf{V'}_1 \backslash \mathbf{V'}_2$.
(i) If $|\mathbf{C'}| \leq |\mathbf{R}|$, and rank deficiency holds, we have $|\mathbf{V'}_1 \cup \mathbf{V'}_2 \cup \mathbf{C'}| = 1 + |\mathbf{C'}| + |\mathbf{A}|$. Therefore, we can detect $\mathbf{C'}$ in $\mathcal{C}$ by the checking function NoCollider, because by removing $\mathbf{C'}$ we have $|\mathbf{V'}_1 \cup \mathbf{V'}_2| = 1 + |\mathbf{A}|$.
(ii) If $|\mathbf{C'}| > |\mathbf{R}|$, and rank deficiency holds when checking $k$, we have $|\mathbf{V'}_1 \cup \mathbf{V'}_2 \cup \mathbf{C'}| = 1 + |\mathbf{R}| + |\mathbf{A}| = k+1$, which means $|\mathbf{V}_1 \cup \mathbf{V}_2| = |\mathbf{V'}_1| + |\mathbf{V'}_2| + |\mathbf{R}| = 1 + |\mathbf{R}| + |\mathbf{A}| + |\mathbf{R}| - |\mathbf{C'}| \leq k$. Therefore, by the unfolding order in Algorithm 2, $\mathbf{C}$ will be taken as the children of $\mathbf{V}_1$ and $\mathbf{V}_2$ first and thus will not induce incorrect rank deficiency. $\square$

Further, under Condition 2, we can show that the correctness of Algorithm 2 will not be influenced even though there might exist colliders in $\mathcal{N}$, which is summarized in the following Lemma.

**Lemma 17** (Under Condition 2, colliders in $\mathcal{N}$ do not harm). *If Condition 2 holds, i.e., for every collider structures* $\mathbf{V}_1$, $\mathbf{V}_2$, $\mathbf{C}$, *and* $\mathbf{A}$, *we have* $|\mathbf{C}| + |\mathbf{A}| \geq |\mathbf{V_1}| + |\mathbf{V_2}|$, *then the correctness of Algorithm 2 will not be influenced by the existence of colliders in* $\mathcal{N}$.

*Proof.* Consider a collider structure $\mathbf{V}_1$, $\mathbf{V}_2$, $\mathbf{C}$, and $\mathbf{A}$. The potential existence of $\mathbf{C}$ in $\mathcal{N}$ will not cause rank deficiency unless it is when $k = |\mathbf{C}| + |\mathbf{A}|$. But under Condition 2, we have $|\mathbf{C}| + |\mathbf{A}| \geq |\mathbf{V_1}| + |\mathbf{V_2}|$, so in Algorithm 2, the colliders $\mathbf{C}$ will be taken as the pure children of $\mathbf{V}_1$ and $\mathbf{V}_2$ first. This holds for every collider structure and thus the correctness of Algorithm 2 will not be influenced. $\square$

Now we are ready to prove Theorem 12, as follows.

---

**Algorithm 2:** Phase1: FindCISkeleton (Stage 1 of PC (Spirtes et al., 2000))

---

**Input** : Samples from $n$ observed variables $\mathbf{X}_{\mathcal{G}}$
**Output:** CI skeleton $\mathcal{G}'$

1   **def** *Stage1PC($\mathbf{X}_{\mathcal{G}}$)***:**
2     Initialize a complete undirected graph $\mathcal{G}'$ on $\mathbf{X}_{\mathcal{G}}$;
3     **repeat**
4       **repeat**
5         Select an ordered pair $\mathsf{X}, \mathsf{Y}$ that are adjacent in $\mathcal{G}'$, s.t., $|\mathrm{Adj}_{\mathcal{G}'}(\mathsf{X})\backslash\{\mathsf{Y}\}| \geq n$;
6         Select a subset $\mathbf{S} \subseteq \mathrm{Adj}_{\mathcal{G}'}(\mathsf{X})\backslash\{\mathsf{Y}\}$ s.t., $|\mathbf{S}| = n$;
7         If $\mathrm{rank}(\Sigma_{\{\mathsf{X}\}\cup\mathbf{C},\{\mathsf{Y}\}\cup\mathbf{C}}) = |\mathbf{C}|$, delete the edge between $\mathsf{X}$ and $\mathsf{Y}$ from $\mathcal{G}'$ and record $\mathbf{S}$ in Sepset$(\mathsf{X}, \mathsf{Y})$ and Sepset$(\mathsf{Y}, \mathsf{X})$.;
8       **until** *all $\mathsf{X}, \mathsf{Y}$ s.t., $|Adj_{\mathcal{G}'}(\mathsf{X})\backslash\{\mathsf{Y}\}| \geq n$ and all $\mathbf{S} \subseteq Adj_{\mathcal{G}'}(\mathsf{X})\backslash\{\mathsf{Y}\}$, $|\mathbf{S}| = n$, tested.*;
9       n:=n+1;
10     **until** *no adjacent $\mathsf{X}, \mathsf{Y}$ s.t., $|Adj_{\mathcal{G}'}(\mathsf{X})\backslash\{\mathsf{Y}\}| < n$*;
11     **return** $\mathcal{G}'$

---

*Proof.* As the existence of colliders in $\mathcal{X}$ will enable more non-overlapping treks, the existence of colliders in $\mathcal{X}$ will not induce incorrect rank deficiency. Taking this and the above two Lemmas 16,17 into consideration, under Condition 1 and 2, the existence of colliders between atomic covers will not influence the correctness of Algorithm 2. During the search process of Phase 2, Lemma 15 allows us to test the rank involving partially observed atomic covers and thus we are able to iteratively find all clusters in $\mathcal{G}$ by making use of Theorem 8, with the direction of some edges undetermined. Further, Theorem 11 allows us to correct clusters induced by the violation of the assumption that when searching $k$ clusters all $k' < k$ clusters have been found, by Phase 3 in Algorithm 4. Therefore, our Algorithm 1 including Phase 1, 2, 3 can identify the Markov equivalence class of $\mathcal{G}$, up to rank invariant operations $\mathcal{O}_{\min}$ and $\mathcal{O}_s$. $\qquad\square$

Corollary 1 is directly from Theorem 12. When there is no latent, the Markov equivalence of $\mathcal{O}_{\min}(\mathcal{O}_s(\mathcal{G}))$ is the Markov equivalence of $\mathcal{G}$ so asymptotically the output of RLCD is the same as that of PC.

### A.19 RANK TEST

We employ canonical correlations (Anderson, 1984) to calculate the rank of covariance matrices. Denote by $\alpha_i$ the $i$-th canonical correlation coefficient between two sets of variables $\mathbf{A}$ and $\mathbf{B}$, under the null hypothesis $\mathtt{rank}(\Sigma_{\mathbf{A},\mathbf{B}}) \leq r$ with $N$ sample size, the statistics $-(N-(p+q+3)/2)\sum_{i=r+1}^{\max(|\mathbf{A}|,|\mathbf{B}|)} \log(1 - \alpha_i^2)$ is approximately $\chi^2$ distributed with $(|\mathbf{A}| - r)(|\mathbf{B}| - r)$ degrees of freedom.

## B   ILLUSTRATIONS OF ALGORITHMS AND MORE DETAILS ABOUT DATASETS

### B.1   EXAMPLES OF GRPAHS THAT EACH METHOD CAN HANDLE

For causal discovery in the presence of latent variables, traditional wisdom often relies on strong graphical constraints for achieving the identifiability of the structure. E.g., Pearl (1988); Zhang (2004) assume that the underlying graph only follows a tree structure; Huang et al. (2022) assumes a more general latent hierarchical structures but edges among observed variables are not allowed; Maeda & Shimizu (2020) allows observed variables to be adjacent but requires that all latent variables are mutually independent.

Illustrative graphs allowed by each method are shown in Figure 11. To be specific, (a) is the illustrative graph allowed by Pearl (1988); Zhang (2004) where each cluster has only one latent variable and observed variables are not allowed to be directly related to each other. (b) is the graph allowed by Huang et al. (2022) where each cluster can have multiple latent variables. However, observed variable cannot be adjacent to each other and observed variables cannot be cause of latent variables. (c) is the graph allowed by Maeda & Shimizu (2020), where all latent variables are required to be

---

**Algorithm 3:** Phase2: FindCausalClusters

---

**Input** : Samples from $n$ observed variables $\mathbf{X}_{\mathcal{G}}$
**Output:** Graph $\mathcal{G}'$

1 **def** *FindCausalClusters($\mathcal{G}'$, $\mathbf{X}_{\mathcal{G}}$):*
2    Active set $\mathcal{S} \leftarrow \mathcal{X}_{\mathcal{G}} = \{\{X_1\}, ..., \{X_n\}\}, k \leftarrow 1$ ;        `// S is a set of covers`
3    **repeat**
4       $\mathcal{G}'$, $\mathcal{S}$, found = Search($\mathcal{G}'$, $\mathcal{S}$, $\mathbf{X}_{\mathcal{G}}$, $k$) ;    `// Only when nothing can be found`
5       If found = 1 then $k \leftarrow 1$ else $k \leftarrow k+1$ ; `// udner current k do we add k by 1`
6    **until** $k$ *is sufficiently large*;
7    **return** $\mathcal{G}'$;
8 **def** *Search($\mathcal{G}'$, $\mathcal{S}$, $\mathbf{X}_{\mathcal{G}}$, $k$):*
9    Rank deficiency set $\mathbb{D} = \{\}$ ;        `// To store rank deficient combinations`
10    **for** $\mathcal{T} \in$ *PowerSet($\mathcal{S}$) (from $\mathcal{S}$ to $\emptyset$)* **do**
11       $\mathcal{S}' \leftarrow (\mathcal{S} \backslash \mathcal{T}) \cup (\cup_{\mathbf{T} \in \mathcal{T}} \text{PCh}_{\mathcal{G}'}(\mathbf{T}))$ ;        `// Unfold S to get S'`
12       **for** $t = k$ *to* $0$ **do**
13          **repeat**
14             Draw a set of $t$ observed covers $\mathcal{X} \subset \mathcal{S}' \cap \mathcal{X}_{\mathcal{G}}$;
15             **repeat**
16                Draw a set of covers $\mathcal{C} \subset \mathcal{S}' \backslash \mathcal{X}$, s.t., $\|\mathcal{C}\| = k - t + 1$ and get $\mathcal{N} \leftarrow \mathcal{S}' \backslash (\mathcal{X} \cup \mathcal{C})$;
17                **if** $rank(\Sigma_{\mathcal{C} \cup \mathcal{X}, \mathcal{N} \cup \mathcal{X}}) = k$ *and NoCollider($\mathcal{C}$, $\mathcal{X}$, $\mathcal{N}$)* **then** Add $\mathcal{C}$ to $\mathbb{D}$ ;
18             **until** *all $\mathcal{C}$ exhausted*;
19             **if** $\mathbb{D} \neq \emptyset$ **then**
20                **for** $\mathcal{D}_i \in \mathbb{D}$ **do**
21                   **if** $|Pa_{\mathcal{G}'}(\mathcal{D}_i) \cup \mathbf{X}| = k$ **then** $\mathbf{P} \leftarrow \text{Pa}_{\mathcal{G}'}(\mathcal{D}_i) \cup \mathbf{X}$ ;
22                   **else** Create new latent variables $\mathbf{L}$, s.t., $\mathbf{P} \leftarrow \mathbf{L} \cup \text{Pa}_{\mathcal{G}'}(\mathcal{D}_i) \cup \mathbf{X}$ and $|\mathbf{L}| = k - |\text{Pa}_{\mathcal{G}'}(\mathcal{D}_i) \cup \mathbf{X}|$ ;
23                   Update $\mathcal{G}'$ by taking elements of $\mathcal{D}_i$ as the pure children of $\mathbf{P}$;
24                   **if** $\mathbf{P}$ *is atomic* **then** Update $\mathcal{S} \leftarrow (\mathcal{S} \backslash \mathcal{D}_i) \cup \mathbf{P}$ ;
25               **return** $\mathcal{G}'$, $\mathcal{S}$, True ;         `// Return to search with k = 1`
26          **until** *all $\mathcal{X}$ exhausted*;
27    **return** $\mathcal{G}'$, $\mathcal{S}$, False ;         `// Return to search with k ← k + 1`

---

**Algorithm 4:** Phase3: RefineCausalClusters

---

**Input** : Graph $\mathcal{G}'$
**Output:** Refined graph $\mathcal{G}'$

1 **def** *RefineCausalCLusters($\mathcal{G}'$, $\mathbf{X}_{\mathcal{G}}$):*
2    **repeat**
3       Draw an atomic cover $\mathbf{V}$ from $\mathcal{G}'$;
4       Delete $\mathbf{V}$, neighbours of $\mathbf{V}$ that are latent, and all relating edges from $\mathcal{G}'$ to get $\hat{\mathcal{G}}$;
5       $\mathcal{G}' = $ FindCausalClusters($\hat{\mathcal{G}}$, $\mathbf{X}_{\mathcal{G}}$);
6    **until** *No more $\mathbf{V}$ found and all $\mathbf{V}$ exhausted*;
7    **return** $\mathcal{G}'$

---

mutually independent. (d) is the graph allowed by the proposed method, where all variables are allowed to be very flexibly related to each other.

### B.2 EXAMPLE OF VIOLATION OF FAITHFULNESS

Below, we provide a special example where faithfulness does not hold. Suppose the true underlying graph $\mathcal{G}$ is $X \xrightarrow{a} Y \xrightarrow{b} Z$ and $X \xrightarrow{c} Z$. If the corresponding SCM is parameterized with $ab + c = 0$, then the faithfulness assumption is violated. This is because, when $ab + c = 0$, there exists another SCM with a graph $\mathcal{G}' : X \xrightarrow{a'} Y \xleftarrow{b'} Z$ that can generate exactly the same observational distribution as that of $\mathcal{G}$, and thus from observational data it is impossible to differentiate $\mathcal{G}$ and $\mathcal{G}'$ (which results in rank($\Sigma_{X,Z}$) = 0). We note that such scenarios are very rare and classical methods like PC (Spirtes et al., 2000) cannot handle these situations either.

---

**Algorithm 5:** Function: NoCollider

---

**Input** : $\mathcal{C}, \mathcal{X}, \mathcal{N}$
**Output:** Whether there exists $\mathbf{O} \in \mathcal{C}$ s.t., $\mathbf{O}$ is a collider of $\mathcal{C}\backslash\{\mathbf{O}\}$ and $\mathcal{N}$

1 **def** *NoCollider($\mathcal{C}, \mathcal{X}, \mathcal{N}$)*:
2    **for** $c = 1$ *to* $|\mathcal{C}| - 1$ **do**
3      Draw $\mathcal{C}' \subset \mathcal{C}$ s.t., $|\mathcal{C}'| = c$;
4      **repeat**
5        **if** $rank(\Sigma_{\mathcal{C}'\cup\mathcal{X},\mathcal{N}\cup\mathcal{X}}) < ||\mathcal{C}' \cup \mathcal{X}||$ **then return** False ;
6      **until** *all $\mathcal{C}'$ exhausted*;
7    **return** True

---

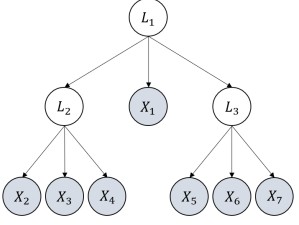

(a) Illustrative graph allowed by Pearl (1988); Zhang (2004).

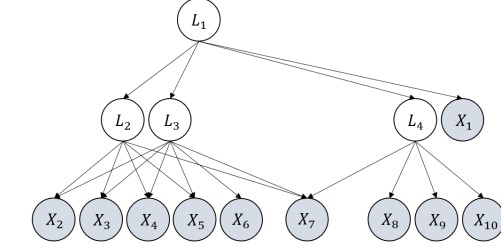

(b) Illustrative graph allowed by Huang et al. (2022).

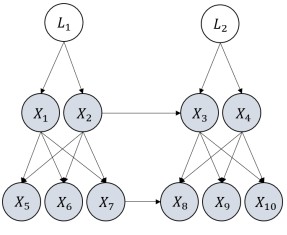

(c) Illustrative graph allowed by Maeda & Shimizu (2020).

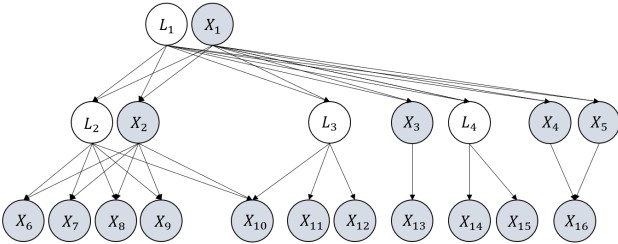

(d) Illustrative graph allowed by the proposed method.

Figure 11: Examples of graphs that are allowed by each method.

### B.3 EXAMPLE OF ATOMIC COVER

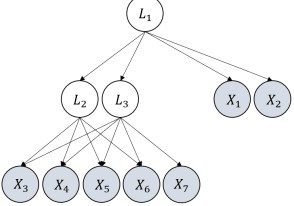

Figure 12: An example graph for showing atomic covers.

**Example 9.** *Take Figure 12 as an example. Here $\{X_1\}$, $\{X_2\}$, $\{X_3\}$, $\{X_4\}$, $\{X_5\}$, $\{X_6\}$ are all atomic covers as they only contain a single observed variable. We also have $\mathbf{V} = \{L_2, L_3\}$ as an atomic cover. To show this, lets check whether conditions (i)-(iii) in Definition 5 are satisfied. For*

*(i), we can let $\mathcal{C} = \{\{X_3\}, \{X_4\}, \{X_5\}\}$, and $\|\mathcal{C}\| = 3 \geq k + 1 - t = 3$. For (ii), we can let $\mathcal{N} = \{\{L_1\}, \{X_6\}, \{X_7\}\}$, and $\|\mathcal{N}\| = 3 \geq k + 1 - t = 3$. For (iii), it can be shown that both $\{L_2\}$ and $\{L_3\}$ cannot be an atomic cover. Therefore, $\{L_2, L_3\}$ is an atomic cover.*

*Next, let's show $\{L_1\}$ is also an atomic cover. As for (i) we can take $\mathcal{C} = \{\{L_2, L_3\}\}$ with $\|\mathcal{C}\| = 2 \geq k - 1 + t = 2$, for (ii) we can take $\mathcal{N} = \{\{X_1\}, \{X_2\}\}$ with $\|\mathcal{N}\| = 2 \geq k - 1 + t = 2$, and (iii) naturally holds as $\{L_1\}$ has a single element. Therefore $\{L_1\}$ is an atomic cover.*

*From the example it is natural to see that when we take an atomic cover (a set) as the unit, we need to use a set of covers, i.e., $\mathcal{C}$, to capture the pure children of a cover.*

### B.4 EXAMPLE FOR PHASE 1

We take Figure 3 (a) as an example. After Phase 1, we will find the CI skeleton $\mathcal{G}'$. In $\mathcal{G}'$, we have three maximal cliques that have cardinality$\geq 3$. They are $\{X_1, X_2, X_3, X_4, X_5, X_6\}$, $\{X_2, X_3, X_7\}$, and $\{X_9, X_{10}, X_{11}, X_{12}\}$. Then we partition them into groups such that two cliques $\mathbf{Q}_1, \mathbf{Q}_2$ are in the same group if $|\mathbf{Q}_1 \cap \mathbf{Q}_2| \geq 2$. Thus we have two groups of cliques $\mathcal{Q}_1 = \{\{X_1, X_2, X_3, X_4, X_5, X_6\}, \{X_2, X_3, X_7\}\}$ and $\mathcal{Q}_2 = \{\{X_9, X_{10}, X_{11}, X_{12}\}\}$. Given $\mathcal{Q}_1$ and $\mathcal{Q}_2$, we get $\mathbf{X}_{\mathcal{Q}_1} = \cup_{\mathbf{Q} \in \mathcal{Q}_1} \mathbf{Q} = \{X_1, X_2, X_3, X_4, X_5, X_6, X_7\}$ and $\mathbf{X}_{\mathcal{Q}_2} = \cup_{\mathbf{Q} \in \mathcal{Q}_2} \mathbf{Q} = \{X_9, X_{10}, X_{11}, X_{12}\}$, the corresponding input to Phase 2 and 3 will be $\mathbf{X}_{\mathcal{Q}_1} \cup \mathbf{N}_{\mathcal{Q}_1} = \{X_1, X_2, X_3, X_4, X_5, X_6, X_7, X_8\}$ and $\mathbf{X}_{\mathcal{Q}_2} \cup \mathbf{N}_{\mathcal{Q}_2} = \{X_9, X_{10}, X_{11}, X_{12}, X_8\}$ respectively.

We next show that if there exist disjoint $\mathbf{A}$, $\mathbf{B}$, $\mathbf{C}$, s.t., (i)-(iv) in Theorem 5 hold, then $\mathbf{A} \cup \mathbf{B} \cup \mathbf{C} \subseteq \mathbf{X}_{\mathcal{Q}}$.

If there exist disjoint $\mathbf{A}$, $\mathbf{B}$, $\mathbf{C}$, s.t., (i)-(iv) in Theorem 5 hold. Then $\mathbf{A}$ itself is a clique, and for all $B \in \mathbf{B}$, $\mathbf{A} \cup \{B\}$ is also a clique. Plus $\mathbf{A}$ and $\mathbf{A} \cup \{B\}$ have at least two common elements as $|\mathbf{A}| \geq 2$. Thus, after our processing, $\mathbf{A}, \mathbf{B}$ will both be subsets of a same $\mathbf{X}_{\mathcal{Q}}$. Plus, by (iv), $\mathbf{C}$ will be a subset of $\mathbf{N}_{\mathcal{Q}}$. Therefore, we have $\mathbf{A} \cup \mathbf{B} \cup \mathbf{C} \subseteq \mathbf{X}_{\mathcal{Q}}$.

### B.5 DETAILED DESCRIPTION AND EXAMPLE FOR PHASE 2

Our search starts with $k = 1$ and an input graph $\mathcal{G}'$ (could be empty) over observed variables. Every time we successfully found rank deficiency with $\texttt{rank} = k$, we update the graph and reset $k$ to 1. On the other hand, if no rank deficiency can be found with current $k$, we add $k$ by 1 (line 5, Alg 3), as we want to ensure all $k'$ clusters, $k' < k$, have been found when searching for $k$, as in Theorem 8.

During the search procedure, we maintain an active set $\mathcal{S}$, which is a set of covers and is initialized as the set of observed covers $\{\{X_1\}, ..., \{X_n\}\}$ (line 2, Alg. 3). We test the rank deficiency over different combinations of $\mathcal{C}$ and $\mathcal{X}$, drawn from $\mathcal{S}'$, where $\mathcal{S}'$ is generated from $\mathcal{S}$ by unfolding some existing clusters (lines 10-11, Alg. 2). Introducing $\mathcal{S}$ and $\mathcal{S}'$ has several merits: (i) When we found a new atomic cover, we want to explore its relation with existing ones. This can be achieved by adding the newly found atomic cover to the active set $\mathcal{S}$ (illustrated in Example 10). (ii) According to Theorem 8, we do not want any descendants of $\mathcal{C}$ to be on the right side of the cross-covariance matrix when testing the rank. This can be achieved by removing all the children of a newly found atomic cover from the active set $\mathcal{S}$. Taking (i) and (ii) together, we update the active set by $\mathcal{S} \leftarrow (\mathcal{S}\backslash\mathcal{C}) \cup \mathbf{P}$ (line 24, Alg. 3). (iii) We unfold $\mathcal{S}$ to get $\mathcal{S}'$, and draw combinations $\mathcal{C}$ and $\mathcal{X}$ from $\mathcal{S}'$ instead of $\mathcal{S}$. This allows the children of existing clusters to re-appear in $\mathcal{C}$ and $\mathcal{X}$, to facilitate finding new clusters that share parents with existing ones, which will be discussed in detail later.

In addition, to establish a unique connection between rank deficiency and atomic covers, we need to avoid colliders and their descendants to appear in $\mathcal{S}'$, as the existence of colliders in $\mathcal{C}$ or $\mathcal{N}$ ($\mathcal{N} \leftarrow \mathcal{S}'\backslash(\mathcal{X} \cup \mathcal{C})$) might induce rank deficiency that does not indicate a correct cluster (example in Appx. B.11). To this end, we take two steps: (i) Every time we found rank deficiency, we further check whether there is a collider in $\mathcal{C}$ (by the *NoCollider* function described in Alg. 5), i.e., check whether there exists $\mathbf{O} \in \mathcal{C}$ s.t., $\mathbf{O}$ is a collider of $\mathcal{C}\backslash\{\mathbf{O}\}$ and $\mathcal{N}$ (line 5 in Alg. 5). If there is a collider, then we ignore the corresponding combination of $\mathcal{C}$ and $\mathcal{X}$ (line 17 in Alg. 3). (ii) We perform unfolding on $\mathcal{S}$ to get $\mathcal{S}'$. That is, every time we consider $\mathcal{T}$, a subset of $\mathcal{S}$, and get $\mathcal{S}' \leftarrow (\mathcal{S}\backslash\mathcal{T}) \cup (\cup_{\mathbf{T} \in \mathcal{T}} \mathrm{PCh}_{\mathcal{G}'}(\mathbf{T}))$ (lines 10-11 in Alg. 3). This allows us to reconsider the pure children of existing atomic covers when choosing combinations of $\mathcal{C}$ and $\mathcal{X}$, and thus, colliders can

be identified (illustrated in Example 10). Taking these two steps together, under the Condition 2, it can be guaranteed that our search procedure will not be affected by the existence of colliders (proof in Appx. A.18).

**Example 10** (Example for Phase 2). *Consider the graph in Figure 4. We start with finding atomic covers with $k = 1$, and we can find that $\{X_3\}$ is a parent of $\{X_8\}$, as in Figure 4(a). At this point, no more $k = 1$ clusters can be found, so next we search for $k = 2$ clusters. Then to identify collider $\{X_7\}$, we only need to consider $\{X_7\}$ and $\{X_8\}$ together as the children of $\{X_2, X_3\}$. After finding such a relationship, we arrive at Figure 4(b), and from now on the collider $\{X_7\}$ will not induce unfavorable rank deficiency anymore (as it is recorded). The next step is to find the relation of $\{X_4\}, \{X_5\}, \{X_6\}$ with $\{L_2, X_2\}$, by taking $\mathcal{X} = \{\{X_2\}\}$ and $\mathcal{C} = \{\{X_4\}, \{X_5\}\}$ or $\{\{X_4\}, \{X_6\}\}$ or $\{\{X_5\}, \{X_6\}\}$, and thus conclude $\{\{X_4\}, \{X_5\}, \{X_6\}\}$ as the pure children of $\{L_2, X_2\}$, as in Fig 4(c). Finally we are able to find the relationship of $\{X_1\}, \{L_2, X_2\}, \{X_3\}$ with $\{L_1\}$, by taking $\mathcal{X} = \{\}$ and $\mathcal{C}$ as $\{\{X_1\}, \{L_2, X_2\}\}, \{\{X_1\}, \{X_3\}\},$ or $\{\{L_2, X_2\}, \{X_3\}\}$, as in Figure 4(d).*

We here give a more detailed example to show the procedure of Phase 2, with the underlying graph $\mathcal{G}$ showed in Figure 4(d).

**Step 1.** Initialize active set $\mathcal{S}$ as $\{\{X_1\}, ..., \{X_8\}\}$, $k$ as 1.

**Step 2.** Get $\mathcal{S}'$ by unfolding $\mathcal{S}$. Currently, $\mathcal{S}' = \{\{X_1\}, ..., \{X_8\}\}$. Now let $k = 1$ and $t = 1$. Draw a set of $t$ observed covers $\mathcal{X} \subset \mathcal{S}' \cap \mathcal{X}_\mathcal{G}$, and draw a set of covers $\mathcal{C} \subset \mathcal{S}' \backslash \mathcal{X}$, s.t., $\|\mathcal{C}\| = k - t + 1$, and check whether rank deficiency holds. We will find that when $\mathcal{X} = \{\{X_3\}\}$ and $\mathcal{C} = \{\{X_8\}\}$, rank deficiency holds and there is no collider detected. Therefore we draw a link from $X_3$ to $X_8$ in $\mathcal{G}'$, as shown in Figure 4(a). Now update the active set $\mathcal{S}$ as $\{\{X_1\}, ..., \{X_7\}\}$.

**Step 3.** Continue searching with $k = 1$, but no more rank deficiency can be found. Therefore, we add $k$ by 1.

**Step 4.** Unfold $\mathcal{S}$ and get $\mathcal{S}' = \{\{X_1\}, ..., \{X_8\}\}$. Now, $k = 2$ and $t = 2$. By drawing $\mathcal{X}$ and $\mathcal{C}$, we will find that when $\mathcal{X} = \{\{X_2\}, \{X_3\}\}$ and $\mathcal{C} = \{\{X_7\}\}$, rank deficiency holds and there is no collider detected. Therefore, we draw links from $X_2X_3$ to $X_7$ in $\mathcal{G}'$, as shown in Figure 4(b). Now update the active set $\mathcal{S}$ as $\{\{X_1\}, ..., \{X_6\}\}$.

**Step 5.** Reset $k = 1$ and search for rank deficiency. No more rank deficiency can be found with $k = 1$, and thus we add $k$ by 1.

**Step 6.** Get $\mathcal{S}'$ by unfolding $\mathcal{S}$, and $\mathcal{S}' = \{\{X_1\}, ..., \{X_6\}\}$. When $k = 2$ and $t = 2$, no more rank deficiency can be found. Therefore, we try $k = 2$ and $t = 1$. By drawing $\mathcal{X}$ and $\mathcal{C}$, we will find that when $\mathcal{X} = \{\{X_2\}\}$ and $\mathcal{C} = \{\{X_4\}, \{X_5\}\}$ or $\{\{X_4\}, \{X_6\}\}$ or $\{\{X_5\}, \{X_6\}\}$, rank deficiency holds and there is no collider detected. Therefore, we conclude that there is an atomic cover. As $k = 2$ but $\|\mathcal{X}\| = 1$, we need one additional latent variable to explain this atomic cover. Thus, we add a new node $L_2$ to $\mathcal{G}'$ (the subscript index for $L$ can be rather arbitrary as long as it is not the same as an existing one), and draw links from $L_2X_2$ to $X_4X_5X_6$ in $\mathcal{G}'$, as shown in Figure 4(c). Now, update the active set $\mathcal{S}$ as $\{\{X_1\}, \{X_2\}, \{X_3\}, \{L_2, X_2\}\}$.

**Step 7.** Reset $k = 1$ and search for rank deficiency. Unfold $\mathcal{S}$ and get $\mathcal{S}' = \{\{X_1\}, \{X_2\}, \{X_3\}, \{L_2, X_2\}\}$. When $k = 1$ and $t = 1$, no more rank deficiency can be found. Therefore we try $k = 1$ and $t = 0$. By drawing $\mathcal{X}$ and $\mathcal{C}$, we will find that when $\mathcal{X} = \{\}$ and $\mathcal{C} = \{\{X_1\}, \{X_2\}\}$ or $\{\{X_2\}, \{X_3\}\}$ or $\{\{X_1\}, \{X_3\}\}$ or $\{\{L_2, X_2\}\}$, rank deficiency holds and there is no collider detected. Therefore, we conclude that there is an atomic cover. All the possible $\mathcal{C}$ will be merged. As $k = 1$ but $\|\mathcal{X}\| = 0$, we need one additional latent variable to explain this atomic cover. Thus, we add a new node $L_1$ to $\mathcal{G}'$, and draw links from $L_1$ to $X_1L_2X_2X_3$ in $\mathcal{G}'$, as shown in Figure 4(d), and update the active set $\mathcal{S}$ as $\{\{L_1\}\}$.

**Step 8.** From now on, no more rank deficiency can be found, and when $k$ is sufficiently large the procedure ends. Output $\mathcal{G}'$, as in Figure 4(d).

### B.6 EXAMPLE FOR PHASE 3

Here, we give an example (see Figure 13) where Phase 2 may result in incorrect latent covers, and thus we need Phase 3 to characterize and refine these incorrect latent covers. Specifically, as in Figure 13), when we look for $k = 3$ clusters, none of the atomic covers $\{L_1\}, \{L_4\}, \{L_2, L_3\}$ has

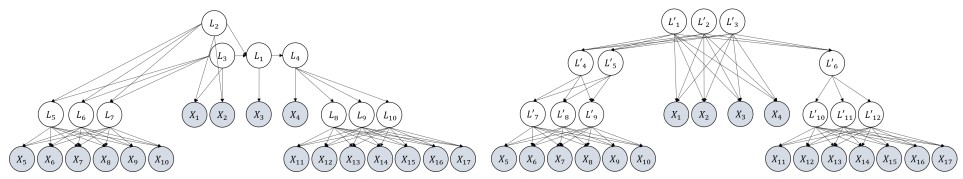

(a) The ground truth graph $\mathcal{G}$.

(b) Algorithm output after phase 2, taken as input of RefineCausalClusters.

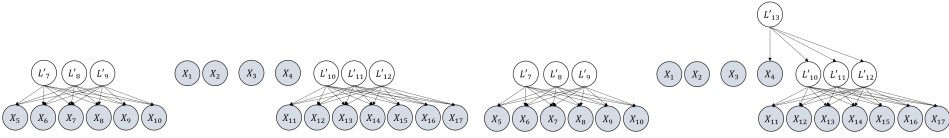

(c) Remove $\{L'_1, L'_2, L'_3\}$ and its neighbours that are latent. Then perform FindCausalClusters.

(d) During FindCausalClusters performed at (c) we first find $\{L'_{13}\}$.

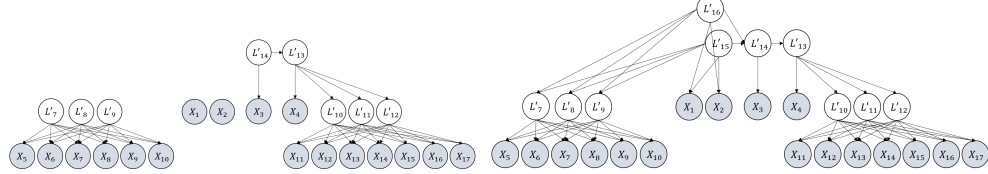

(e) During FindCausalClusters performed at (c), we then find $\{L'_{14}\}$.

(f) During FindCausalClusters performed at (c), we finally find $\{L'_{15}, L'_{16}\}$.

Figure 13: Example in subfigure (a) is the real graph $\mathcal{G}$. After phase 2 the output graph still contains a fake cluster, as in (b). After phase 3, the output graph will be correct, as in (f).

been discovered. Therefore, in Phase 2, when looking for $k = 3$ clusters, we will find a combination of $\mathcal{C} = \{\{X_1\}, \{X_2\}, \{X_3\}, \{X_4\}\}$ and $\mathcal{X} = \{\}$ that causes rank deficiency, and thus we will mistakenly create an atomic cover $\{L'_1, L'_2, L'_3\}$ with their pure children $\{X_1\}, \{X_2\}, \{X_3\}, \{X_4\}$, as in Figure 13(b).

Fortunately, this incorrect cluster will not affect the identification of other clusters in the graph: e.g., in Figure 13(b), the covers $\{L'_7, L'_8, L'_9\}, \{L'_{10}, L'_{11}, L'_{12}\}$ are correctly found, except that the neighbors of the wrong atomic cover $\{L'_1, L'_2, L'_3\}$ could be incorrect. This allows us to take a further look into the incorrect cluster and refine it based on Theorem 11 (the proof of which is in Appendix A.17).

As shown in Figure 13, the subfigure (a) is the underlying graph $\mathcal{G}$. After phase 2 the output graph $\mathcal{G}'$ in (b) contains incorrect cover $\mathbf{V} = \{L'_1, L'_2, L'_3\}$. In (c), we first calculate $\hat{\mathcal{G}}$, which is got by deleting $\mathbf{V}$, all neighbours of $\mathbf{V}$ that are latent, and all relating edges of them from $\mathcal{G}'$. The resulting $\hat{\mathcal{G}}$ is shown in Figure 13 (c). After that, we perform FindCausalClusters($\hat{\mathcal{G}}, \mathbf{X}$), and then the clusters $\{X_1, X_2\}, \{X_3\}$, and $\{X_4\}$ can be correctly found, as shown in Figure 13 (d)(e)(f).

## B.7 GRAPH EXAMPLES WITH VARIABLES ALL OBSERVED

Please refer to Figure 15.

## B.8 GRAPH EXAMPLES FOR LATENT TREE MODELS

Please refer to Figure 18.

## B.9 GRAPH EXAMPLES FOR LATENT MEASUREMENT MODELS

Please refer to Figure 17.

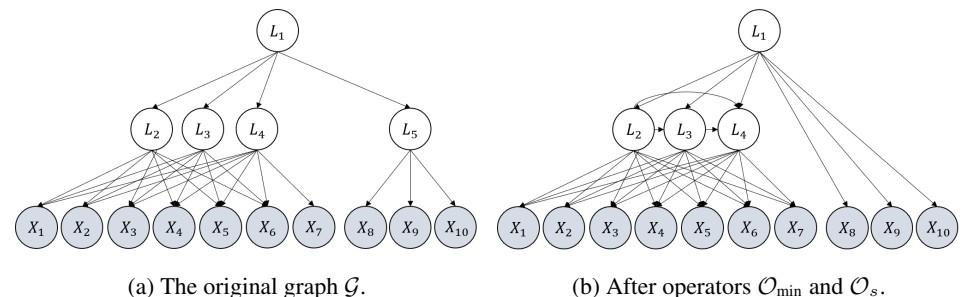

(a) The original graph $\mathcal{G}$.        (b) After operators $\mathcal{O}_{\min}$ and $\mathcal{O}_s$.

Figure 14: Example to show graph operators $\mathcal{O}_{\min}$ and $\mathcal{O}_s$.

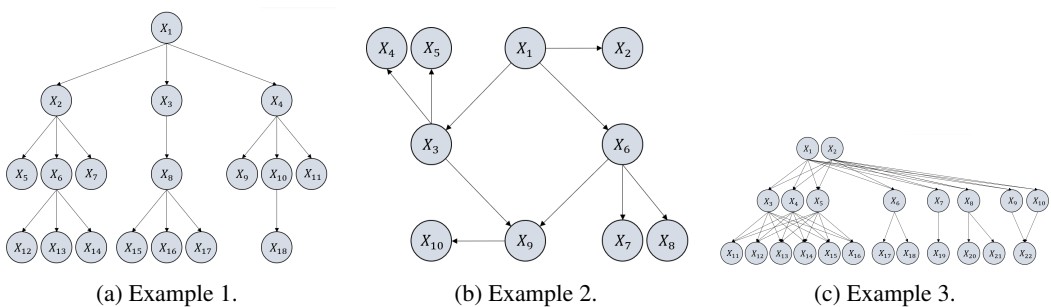

(a) Example 1.        (b) Example 2.        (c) Example 3.

Figure 15: Examples of graphs that have only observed variables.

## B.10   GRAPH EXAMPLES FOR GENERAL LATENT MODELS

Please refer to Figure 19.

## B.11   ILLUSTRATIVE EXAMPLE OF CONSIDERING COLLIDERS IN PHASE 2

For example, in Figure 16, suppose that we have already found the cover $L_1$ as the parent of cluster $X_1 X_2$, and $L_2$ as the parent of cluster $X_6 X_7$. Next, we search for $k = 2$ clusters and take $\mathcal{C} = \{\{X_3\}, \{X_4\}, \{L_2\}\}$ and $\mathcal{X} = \{\}$, and then we have rank deficiency $\text{rank}(\Sigma_{\mathcal{C} \cup \mathcal{X}, \mathcal{N} \cup \mathcal{X}}) = 2$. However, this rank deficiency does not imply a correct cluster as there is a set of collider $\{L_2\}$ inside $\mathcal{C}$. Fortunately, it can be detected by Algorithm 5. Specifically, if we take $\mathcal{C}' = \{\{X_3\}, \{X_4\}\}$, we can find that $\text{rank}(\Sigma_{\mathcal{C}' \cup \mathcal{X}, \mathcal{N} \cup \mathcal{X}}) = \text{rank}(\Sigma_{X_3 X_4, X_1 X_2 X_5 X_6}) = 1$ (line 5 in Algorithm 5), which means there exists a smaller group of rank deficiency caused by removing the collider in $\mathcal{C}$. Thus, we conclude that $\mathcal{C} = \{\{X_3\}, \{X_4\}, \{L_2\}\}$ and $\mathcal{X} = \{\}$ is not a correct combination and will not consider them for forming a cluster (as in line 1 in Algorithm 2).

## B.12   EXAMPLES FOR GRAPH OPERATORS

Suppose a graph $\mathcal{G}$ of a latent linear model in Figure 14(a) is $\mathcal{G}$. After applying $\mathcal{O}_{\min}(\mathcal{O}_s(\mathcal{G}))$, we have the graph in Figure 14(b). Specifically, the $\mathcal{O}_s$ operator adds the following edges: $L_2 \to X_7$, $L_3 \to X_7$, $L_2 \to L_3$, $L_2 \to L_4$, and $L_3 \to L_4$. The $\mathcal{O}_{\min}$ operator delete $L_5$ and add an edge from $L_1$ directly to $X_8$, $X_9$, and $X_{10}$. For $\mathcal{G}$, such two operators will not change the rank in the infinite sample case.

## B.13   GRAPHICAL RELATIONS BETWEEN COVERS AND SET OF COVERS

The relation between covers naturally follows the relation between a set of variables. For example, in Figure 13(a), the pure children of $\{L_4, L_5\}$ is $\{X_5, X_6, X_7, X_8\}$. For the relation between sets of covers, it also follows the relationship between variables. E.g., in Figure 13(a), the parents of $\{\{L_4\}, \{L_5\}\}$ is a set of nodes $\{L_1\}$.

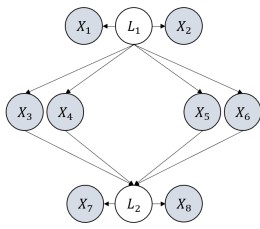

Figure 16: Example of checking colliders in $\mathcal{N}$.

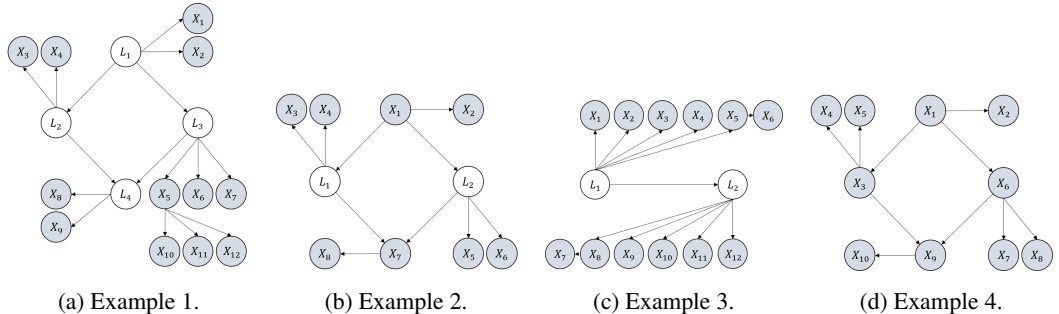

| (a) Example 1. | (b) Example 2. | (c) Example 3. | (d) Example 4. |
|---|---|---|---|

Figure 17: Examples of Latent Measurement Graphs.

## B.14 DISCUSSIONS ON CHECKING COLLIDERS COMPLETELY

With our search procedure that checks colliders in Algorithm 5, we can make sure that the existence of colliders between atomic covers in $\mathcal{C}$ will not induce incorrect clustering results. However, we note that there are still chances that colliders are in $\mathcal{N}$. If Condition 2 holds, then we can make sure that the existence of colliders in $\mathcal{N}$ will not induce fake clusters. In fact, there is a way to further check whether there exist colliders in $\mathcal{N}$. Specifically, in the line 17 of Algorithm 2, if we have $\text{rank}(\Sigma_{\mathcal{C} \cup \mathcal{X}, \mathcal{N} \cup \mathcal{X}}) = k$, and $\text{NoCollider}(\mathcal{C}, \mathcal{X}, \mathcal{N})$ returns True, we can further check whether there exist a set of covers $\mathcal{N}' \subseteq \mathcal{N}$ such that $\mathcal{N}'$ consists of all the colliders between $\mathcal{C}$ and $\mathcal{N} \backslash \mathcal{N}'$. To this end, we just enumerate all the possible subsets $\mathcal{N}'$ of $\mathcal{N}$. If $\mathcal{N}'$ is the set of all the colliders, then it must be that (i) $\text{rank}(\Sigma_{\mathcal{C} \cup \mathcal{X}, (\mathcal{N} \backslash \mathcal{N}') \cup \mathcal{X}}) = k' < k$, and (ii) $\text{rank}(\Sigma_{\mathcal{C} \cup \mathcal{X} \cup \mathcal{N}', (\mathcal{N}) \cup \mathcal{X}}) > k' + ||\mathcal{N}'||$.

Take Figure 16 as an example. First, we check whether Condition 2 holds. As $|\mathbf{C}| + |\mathbf{A}| = |\{L_1\}| + |\{L_2\}| = 2 < |\mathbf{V_1}| + |\mathbf{V_2}| = |\{X_3, X_4\}| + |\{X_5, X_6\}| = 4$, Condition 2 does not hold. Therefore, when checking $k = 2$, if we take $\mathcal{X} = \{\}$, $\mathcal{C} = \{\{X_3\}, \{X_4\}, \{X_5\}\}$, and $\mathcal{N} = \{\{X_1\}, \{X_2\}, \{X_6\}, \{X_7\}, \{X_8\}\}$ in line 17 of Algorithm 2, we will find that $\text{rank}(\Sigma_{\mathcal{C} \cup \mathcal{X}, \mathcal{N} \cup \mathcal{X}}) = k = 2$, which implies an incorrect cluster as the cardinality of parents of $\{\{X_3\}, \{X_4\}, \{X_5\}\}$ should be only 1. Fortunately, in this scenario, we can detect that $\mathcal{N}' = \{\{X_7\}, \{X_8\}\} \subseteq \mathcal{N}$ is the set of all the colliders, by finding that (i) $\text{rank}(\Sigma_{\mathcal{C} \cup \mathcal{X}, (\mathcal{N} \backslash \mathcal{N}') \cup \mathcal{X}}) = 1 < k = 2$, and (ii) $\text{rank}(\Sigma_{\mathcal{C} \cup \mathcal{X} \cup \mathcal{N}', (\mathcal{N}) \cup \mathcal{X}}) = 4 > 1 + ||\mathcal{N}'|| = 3$.

As mentioned in Section 5, by adding this check function to our algorithm (specifically to line 17 in Algorithm 2 before adding $\mathcal{C}$ to $\mathbb{D}$), we can achieve better identifiability that relies on Condition 1 only. However, that additional checking function is computationally inefficient.

## B.15 EVALUATION METRIC DETAILS

The definition of F1 is as follows. $\text{F1} = \frac{2 * \text{Recall} * \text{Precision}}{\text{Recall} + \text{Precision}}$, $\text{Recall} = \frac{\text{TP}}{\text{TP} + \text{FN}}$, and $\text{Precision} = \frac{\text{TP}}{\text{TP} + \text{FP}}$, where TP, FP, and FN denote True Positive, False Positive, and False Negative, respectively.

For a fair comparison, we need to align the latent variables in the output graph $\mathcal{G}'$ of a method with the latent variables in the ground truth graph $\mathcal{G}$. To this end, we first pad each result by adding latents that have no edge to any other variables to match the number of latents in the ground truth graph. On the other hand, if the number of latents is more than that of the ground truth $\mathcal{G}$, all different combinations will be tried. Finally, we try all different permutations of latent variables to test the

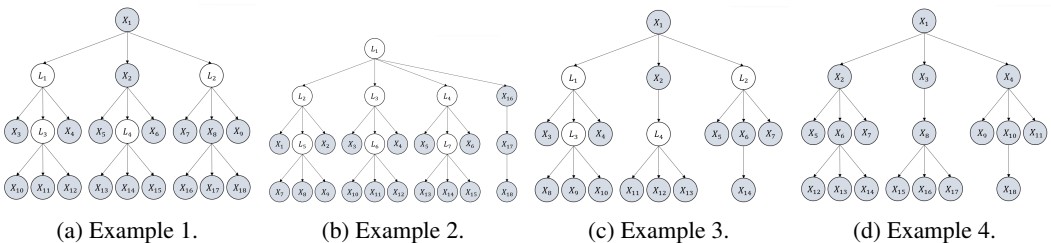

Figure 18: Examples of Latent Tree Graphs.

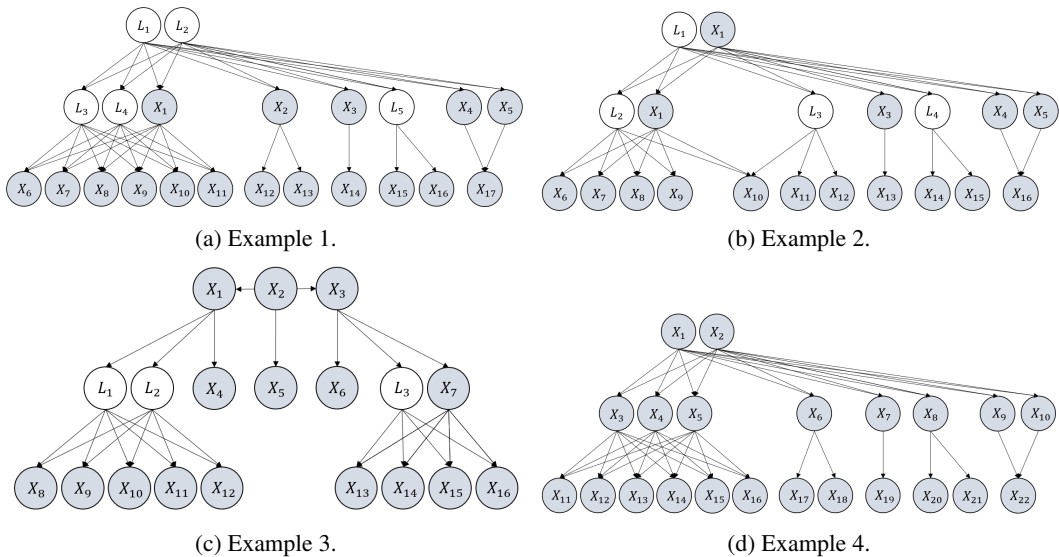

Figure 19: Examples of Latent General Graphs.

F1 score. For each method, the final F1 score is taken as the best F1 score among all possible combinations and permutations.

## B.16 MORE DETAILS OF EXPERIMENTS ON SYNTHETIC DATA

Our code is implemented with Python 3.7. Asymptotically speaking, if the ground gruth graph is a DAG, then there will be no cycle in our result. However, in the finite sample case, rank test results could be self-contradictary. Therefore in our implementation we explictly prevent that by checking whether cycles may occur every time before concluding a cluster. As different methods employ different statistical tests that may perform differently, the hyperparameter $\alpha$ is chosen from $\{0.1, 0.05, 0.01, 0.005\}$ in favor of each method to ensure their best performance and thus a fair comparison. For the proposed method we employ $\alpha = 0.005$ for the procedure of finding latent variables, while for the first stage we empirically find that using a rather big $\alpha$ would be better. This is because the first stage of PC is good at deleting edges and thus bad at recall, and the following procedure would expects input with high recall rather than high precision. We conduct all the experiments with single Intel(R) Xeon(R) CPU E5-2470. Our proposed method and GIN (Xie et al., 2020) take around 3 hours to finish all the experiments (three random seeds and three different sample sizes), and Hier. rank (Huang et al., 2022) takes around 1 hour. PC (Spirtes et al., 2000) and FCI (Spirtes et al., 2013) take around 10 minutes, while RCD (Maeda & Shimizu, 2020) takes around two days to finish the experiments. For GIN, RCD, and Hier. rank, we employ their original implementation while for PC and FCI we use the causal-learn python package https://causal-learn.readthedocs.io/en/latest/.

The time complexity of our proposed algorithm is upper bounded by $\mathcal{O}(l \sum_{k=1}^{K} \sum_{t=0}^{k} \binom{n}{t} \binom{n-t}{k+1-t})$, where $n$ is the number of measured variables, $K$ is the cardinality of the largest cover of the estimated graph, with $K \ll n$, and $l$ is the number of levels of the estimated graph, with $l \ll n$.

It is also possible to exhaustively enumerate all possible graphs and check whether one of them may be aligned with observational rank information from data. However, that would be very computationally expensive. Assume that the underlying graph consists of $n$ measured variables and $m$ latent variables. To conduct an exhaustive search for causal clusters, we need to enumerate all possible numbers of latent vars and then enumerate all possible structures, which results in an approximate total of $\sum_{i=1}^{m} 3^{(i+n)(i+n-1)/2}$ possible combinations. In our synthetic data, which comprises an average of 15 measured variables and 4 latent variables, the mere act of enumeration already demands $10^{66}$ seconds (in our Python environment), which is computationally unacceptable.

## B.17 DETAILED INFORMATION OF THE BIG FIVE PERSONALITY DATASET

Data was collected through an interactive online personality test `https://openpsychometrics.org/`. Participants were informed that their responses would be recorded and used for research at the beginning of the test and asked to confirm their consent at the end of the test. Items were rated on a five-point scale: 1=Disagree, 2=Slightly disagree, 3=Neutral, 4=Slightly agree, 5=Agree (0=missed). Datapoints with missing values have been filtered out. Some additional information is also collected including Race, Age, and Gender but are not used in our experiment. The Markov equivalence class of Figure 5 is generated by using our proposed method, while we further apply GIN (Xie et al., 2020) to determine directions between latent variables. The five personality dimensions are Openness, Conscientiousness, Extraversion, Agreeableness, and Neuroticism (O-C-E-A-N). Below are the raw questions. E.g., E1 denotes the first question for the Extraversion score.

E1 I am the life of the party.
E2 I don't talk a lot.
E3 I feel comfortable around people.
E4 I keep in the background.
E5 I start conversations.
E6 I have little to say.
E7 I talk to a lot of different people at parties.
E8 I don't like to draw attention to myself.
E9 I don't mind being the center of attention.
E10 I am quiet around strangers.
N1 I get stressed out easily.
N2 I am relaxed most of the time.
N3 I worry about things.
N4 I seldom feel blue.
N5 I am easily disturbed.
N6 I get upset easily.
N7 I change my mood a lot.
N8 I have frequent mood swings.
N9 I get irritated easily.
N10 I often feel blue.
A1 I feel little concern for others.
A2 I am interested in people.
A3 I insult people.
A4 I sympathize with others' feelings.
A5 I am not interested in other people's problems.
A6 I have a soft heart.
A7 I am not really interested in others.
A8 I take time out for others.
A9 I feel others' emotions.
A10 I make people feel at ease.
C1 I am always prepared.
C2 I leave my belongings around.
C3 I pay attention to details.
C4 I make a mess of things.
C5 I get chores done right away.
C6 I often forget to put things back in their proper place.

C7 I like order.
C8 I shirk my duties.
C9 I follow a schedule.
C10 I am exacting in my work.
O1 I have a rich vocabulary.
O2 I have difficulty understanding abstract ideas.
O3 I have a vivid imagination.
O4 I am not interested in abstract ideas.
O5 I have excellent ideas.
O6 I do not have a good imagination.
O7 I am quick to understand things.
O8 I use difficult words.
O9 I spend time reflecting on things.
O10 I am full of ideas.

### B.18    MORE ANALYSIS OF THE RESULTS FOR THE BIG FIVE

A prevalent theory of personality is that personality dimensions (factors or traits) are latent causes of the responses to personality inventory items, which are indicators of the latent construct. For instance, extraversion yields high scores for the indicators "I like to go to parties" and "I like people." Thus, the responses to inventory items are outcomes of one's position on the latent dimension. However, there is also the suggestion of a network perspective in which personality structure is viewed in terms of microcausal connections in a complex network (Wright, 2017) and personality dimensions emerge out of the connectivity structure (Cramer et al., 2012). This is a radical divergence from the conventional viewpoint that dimensions are causes of the relevant indicators. For instance, a network would show that instead of being two distinct markers of extraversion, one might say "I like to go to parties" because "I like people" (Cramer et al., 2012); or in the case of openness, "I am full of ideas" because "I have a vivid imagination". Our result in Figure 5 indicates, however, that our method adheres to a network perspective while identifying groups of closely connected items that are predictable under a latent dimension model. Further, it can be observed that causal links occur between latent dimensions, between observed indicators, and among latents and indicators. Following are interesting aspects of our results.

(i) L1, L2, L3 and L5. While L1, L2, and L3 clearly delineate conscientiousness, agreeableness, and extraversion as causes of the corresponding item responses, it is different in the case of openness (L5). Openness to experience can lead to excellent ideas brought about by active imagination, reflection, and understanding of things. Moreover, those who develop a rich vocabulary will have the propensity to think critically and read more, behaviors that also birth ideas.

(ii) L1→L6→L3. Conscientiousness and openness are most frequently associated with achievement (Gatzka, 2021). In our results, those who are organized, efficient at tasks, thorough, systematic, or exacting will comprehend things quickly, demonstrate sophistication in language, or are good at introspection. These could instill a sense of confidence and assurance that encourages assertive, verbal, and bold behaviors, among other extraversion markers.

(iii) L1→L2→L3. People who score highly on conscientiousness are frequently perceived as perfectionists, high achievers, overly focused on personal goals, preoccupied with flawless task execution, overly demanding, and headstrong (Le et al., 2011)(Curşeu et al., 2019). Our findings suggest that due to such behaviors, highly conscientious individuals would judge other people on their accomplishments and results, without giving consideration to others' feelings. Consequently, they act distant, uncommunicative, or unsociable. These can be reasons why they have been found not to engage in group behaviors that lead to straining relationships and tend to take criticism poorly (Curşeu et al., 2019), as well as refrain from conversing about interpersonal issues because they have no bearing on achieving task objectives (Curşeu et al., 2019). On the other hand, those who are both highly conscientious and agreeable put the needs of others before their own (Lord, 2007), at times to the point of pleasing others by overlooking their mistakes, doing things for others because they cannot say no, not disclosing performance gaps and withholding dissident opinions out of aversion to conflict and a lack of competitiveness (Graziano & Tobin, 2002)(Howard & Howard, 2010)(Curşeu et al., 2019). Thus, they will resolve problems on their own, not to draw attention to themselves,

have little to say, or would rather stay in the background in order to be integrated. Those who score low on agreeableness are perceived to be unempathetic, unfriendly, and untrustworthy, consequently leading to introvertive behaviors as well.

**(iv) L1 and L3 together as common causes.** Conscientious individuals who care about being liked by others despite being focused, detail-oriented, and exacting, will make efforts to make people feel at ease amid these behaviors. Otherwise, conscientious individuals who care not about what others think of them could be quick to lambast others if they perform poorly. Low-conscientious people, those who are disorganized, messy, sloppy, and negligent, will also tend to make people at ease in order to remain in their good graces.

**(v) No latent variable for neuroticism.** Our method did not discover any latent variable that is supposed to correspond to neuroticism. One possible interpretation would be that the question "[N10]: I often feel blue." is designed so well that it fully captured the sense of neuroticism.

**(vi) Responses to indicators influence other responses.** It is the question items, not the latent dimensions, that can be perceived to have caused the succeeding responses, as in the case of N10→N8→N7, N10→O9, O2→O4, and O1→O8, all of which are plausible. Mood swings are common with depression, and frequent mood swings cause emotions to fluctuate rapidly and intensely, switching between positive and negative emotions. Some people may find themselves reflecting a lot on things because they are trying to figure out what makes them frequently feel blue and how to cope with it. A person who has trouble understanding abstract concepts is unlikely to be particularly interested in them. Finally, one who has a rich vocabulary will not be constrained from using unusual words that are hard to comprehend.

## C  RELATED WORK AND BROADER IMPACTS

### C.1  RELATED WORK

Causal discovery aims to identify causal relationships from observational data. Most existing approaches are based on the assumption that there are no latent confounders (Spirtes et al., 2000; Chickering, 2002b; Shimizu et al., 2006a; Hoyer et al., 2009; Zhang & Hyvärinen, 2009), and yet this assumption barely holds for real-life problems. Thus, causal discovery methods that can handle the existence of latent variables are crucial. Existing causal discovery methods for handling latent variables can be categorized into the following folds.

**(i) Conditional independence constraints.** The FCI algorithm (Spirtes et al., 2000) and its variants (Colombo et al., 2012; Pearl, 2000; Akbari et al., 2021). This line of work checks conditional independence over observed variables to identify the causal structure over observed variables up to a maximal ancestral graph. They can deal with both linear and nonlinear causal relationships, but there are large indeterminacies in their results, e.g., the existence of an edge and confounders. Plus, they cannot consider causal relationships between latent variables. Based on CI tests, Triantafillou & Tsamardinos (2015) proposes a method that can co-analyze multiple datasets that share common variables and sort the significance tests to address conflicts from statistical errors. **(ii) Tetrad condition.** This line of work makes use of the rank constraints of every $2 \times 2$ off-diagonal sub-covariance matrix to locate latent variables and thus find the causal skeleton based on linear relationships between variables (Silva et al., 2006; Kummerfeld & Ramsey, 2016; Wang, 2020; Pearl, 1988). One limitation of this line of work is that they assume each measured variable is influenced by only one latent parent, and each latent variable must have more than three pure measured children. **(iii) Matrix decomposition.** This line of work proposes to decompose the precision matrix into a low-rank matrix and a sparse matrix, where the former represents the causal structure from latent variables to measured variables and the latter represents the causal structure over measured variables, under certain assumptions (Chandrasekaran et al., 2011; 2012; Anandkumar et al., 2013). E.g., Anandkumar et al. (2013) decomposed the covariance matrix into a low-rank matrix and a diagonal matrix, by assuming three times more measured variables than latent variables. **(iv) Over-complete independent component analysis (ICA).** Over-complete ICA allows more source signals than observed signals, and thus can be used to learn the causal structure with latent variables (Shimizu et al., 2009), and yet they normally do not consider the causal structure among latent variables. The estimation of over-complete ICA models could be hard to reach global optimum without further assumptions (Entner & Hoyer, 2010; Tashiro et al., 2014). **(v) Generalized independent noise (GIN).** The GIN

Table 4: Structural Hamming Distance (SHD) of compared methods on different types of latent graphs where the values are averaged over three random seeds. **The smaller the better.**

| Algorithm | | **SHD for skeleton among all variables $\mathbf{V}_{\mathcal{G}}$ (both $\mathbf{X}_{\mathcal{G}}$ and $\mathbf{L}_{\mathcal{G}}$)** | | | | | |
|---|---|---|---|---|---|---|---|
| | | **Ours** | Hier. rank | PC | FCI | GIN | RCD |
| *Latent+tree* | 2k | **6.9** | 9.3 | 23.7 | 23.7 | 20.5 | 22.2 |
| | 5k | **3.2** | 9.0 | 25.0 | 24.3 | 21.2 | 23.5 |
| | 10k | **0.7** | 9.0 | 25.1 | 24.3 | 20.0 | 24.0 |
| *Latent+measm* | 2k | **4.6** | 8.1 | 14.3 | 14.7 | 10.8 | 15.4 |
| | 5k | **3.8** | 7.7 | 15.0 | 15.0 | 9.2 | 16.2 |
| | 10k | **2.9** | 7.4 | 15.5 | 14.8 | 9.2 | 16.0 |
| *Latent general* | 2k | **27.1** | 28.1 | 38.0 | 37.6 | 36.4 | 36.5 |
| | 5k | **23.0** | 26.0 | 38.2 | 36.8 | 33.8 | 32.5 |
| | 10k | **21.4** | 26.0 | 39.0 | 37.1 | 34.1 | 36.1 |

condition is an extension of the independent noise condition when latent variables exist. Based on non-gaussianity it leverages higher-order statistics to identify latent structures. E.g., Xie et al. (2020) allows multiple latent parents behind every pair of observed variables and can identify causal directions among latent variables, and yet it requires at least twice measured children as latent variables. Dai et al. (2022) proposes a transformed version of GIN to handle measurement errors. **(vi) Mixture oracles-based.** Kivva et al. (2021) proposes a mixture oracles-based method to identify the causal structure in the presence of latent variables where the causal relationships can be nonlinear. It is based on assumptions that the latent variables are discrete and each latent variable has measured variables as children. **(vii) Rank deficiency.** Recently Huang et al. (2022) proposes to leverage rank deficiency of sub-covariance of observed variables to find the underlying causal structure in the presence of latent variables. Our method differs in that we consider a more general setting, i.e., we allow hidden variables can be causally related to each other, form a hierarchical structure (i.e., the children of hidden variables can still be hidden), and even serve as both confounders and intermediate variables for observed variables. Our graphical condition for identifiability also generalizes the condition in (Huang et al., 2022) to cases where edges between observed variables are allowed. **(viii) Heterogeneous data.** Huang* et al. (2020) considere a special type of latent confounders that can be represented as a function of domain index or a smooth function of time. This line of work makes use of domain index or time index as a surrogate to remove confounders' influence and consequently identify causal structure over observed variables. **(ix) Score based.** Agrawal et al. (2021) propose a score-based method for latent variable causal discovery, by assuming additional structure among latent confounders.

The most related work to our method is Hier. Rank Huang et al. (2022). Compared to Hier. Rank, our graphical conditions are not only strictly but also much weaker. Our conditions are strictly weaker in the sense that Hier. Rank can be taken as a special case of the proposed method by disallowing direct edges between observed variables, which is formally captured by our Corollary 1. Our conditions are much weaker in the sense that, basically we allow latent variables and observed variables to be flexibly related and exist everywhere in a graph, which is illustrated in Figure 11 (b) v.s. Figure 11 (d). The reason why we are able to identify these latent structures that Hier. Rank cannot identify, lies in that we utilize the rank constraints in a more flexible and comprehensive fashion. Specifically, Hier. Rank only uses the part of the rank information rank($\Sigma_{\mathbf{A},\mathbf{B}}$) where $\mathbf{A} \cap \mathbf{B} = \emptyset$, while the proposed method uses the rank($\Sigma_{\mathbf{A},\mathbf{B}}$) where $\mathbf{A}$ and $\mathbf{B}$ are rather arbitrary and thus more t-separations can be inferred. The extra graphical information allows us to make use of, e.g., Lemma 10 for identifying edges between observed variables, and Theorem 8 to identify atomic covers that are partially hidden partially observed.

# D  ADDITIONAL INFORMATION

## D.1  EMPIRICAL RESULT USING SHD

In this section we further show the performance of each method using the Structural Hamming Distance (SHD). As shown in Table 4, The SHD of the proposed RLCD method to the ground truth is consistently smaller than all comparative methods under all the settings, which again validates RLCD in the finite sample cases.

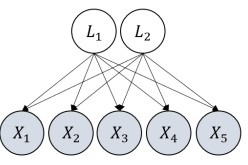 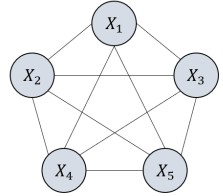

(a) In $\mathcal{G}_1$, Condition 1 is not satisfied, in the sense that $\mathsf{L}_1\mathsf{L}_2$ do not have enough pure children and neighbours.

(b) Given $\mathcal{G}_1$, RLCD outputs $\mathcal{G}_1'$, which is not informative but correct as they are rank-equivalent.

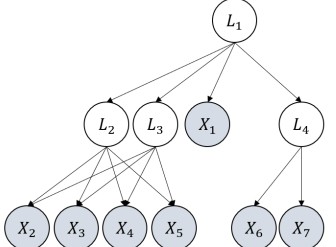 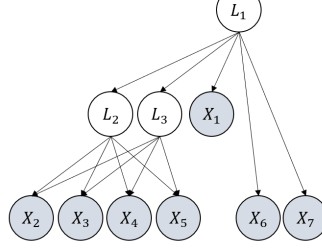

(c) In $\mathcal{G}_2$, Condition 1 is not satisfied, in the sense that $\mathsf{L}_4$ does not have enough pure children and neighbours.

(d) Given $\mathcal{G}_2$, RLCD outputs $\mathcal{G}_2'$, which is correct and informative, though not exactly the same as $\mathcal{G}_2$.

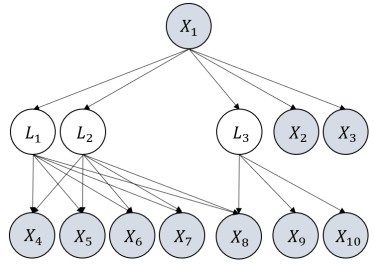 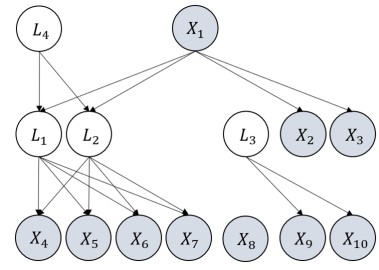

(e) In $\mathcal{G}_3$, the Condition 2 is not satisfied, because $|\{\mathsf{L}_1, \mathsf{L}_2\}| + |\{\mathsf{L}_3\}| > |\{\mathsf{X}_1\}| + |\{\mathsf{X}_8\}|$.

(f) Given $\mathcal{G}_3$, RLCD outputs $\mathcal{G}_3'$, which is incorrect but we can infer from the result that conditions are violated.

Figure 20: Examples to show that even when graphical conditions are not satisfied, the proposed RLCD can provide the correct result (though may be uninformative), or infer that the condition is violated.

## D.2 VIOLATION OF GRAPHICAL CONDITIONS

In this section, we shall discuss what would the output of the proposed method be when graphical conditions 1 2 are not satisfied.

(i) The result is correct but uninformative. For instance, in Figure 20 (a), $\mathsf{L}_1\mathsf{L}_2$ do not belong to any atomic cover, as they do not have enough pure children plus neighbours. Thus, Condition 1 is not satisfied. In this scenario, though the output of the proposed method is not informative, the result is correct. It is not informative in the sense that the result in Figure 20 (b) fails to inform us the existence of latent variables. The result is correct in the sense that it correctly outputs the CI skeleton, which is rank-equivalent to the ground truth $\mathcal{G}_1$. In other words, the output graph $\mathcal{G}_1'$ and the ground truth $\mathcal{G}_1$ are able to entail the same set of observational rank constraints and no algorithm can differentiate them solely by rank information.

(ii) The result is correct and informative, though it uses a more compact graph as the representation of the rank-equivalence class. An example is given in Figure 20 (c), where $\mathsf{L}_4$ does not belong to any atomic cover (as no enough pure children plus neighbours), and thus $\mathcal{G}_2$ does not satisfy

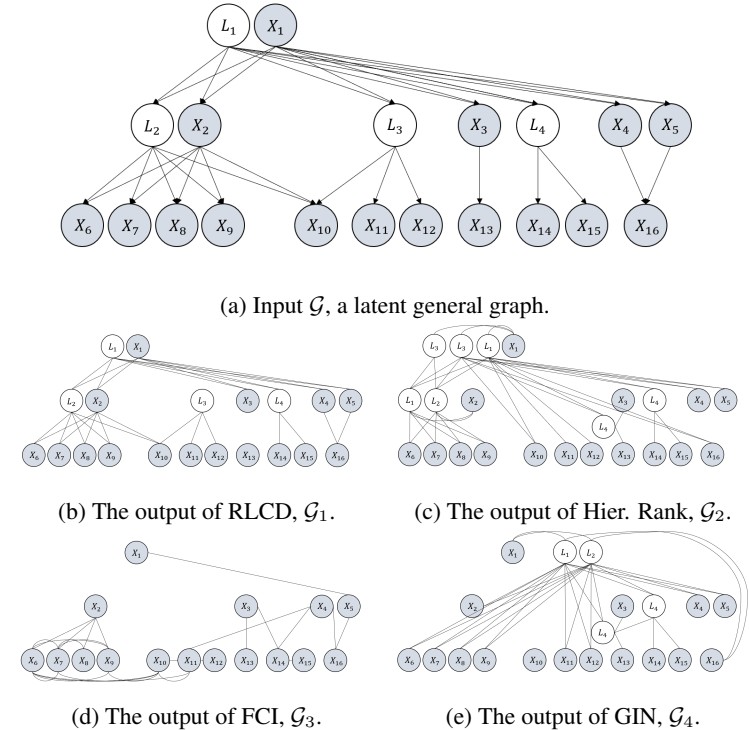

(a) Input $\mathcal{G}$, a latent general graph.

(b) The output of RLCD, $\mathcal{G}_1$.  (c) The output of Hier. Rank, $\mathcal{G}_2$.

(d) The output of FCI, $\mathcal{G}_3$.  (e) The output of GIN, $\mathcal{G}_4$.

Figure 21: Illustrative examples of the output of each method given data generated from $\mathcal{G}$. We only show the skeleton here for simplicity.

Condition 1. However, the proposed RLCD still outputs the correct and informative result $\mathcal{G}_2'$. The output $\mathcal{G}_2'$ is correct in the sense that $\mathcal{G}_2'$ and $\mathcal{G}_2$ are rank-equivalent, i.e., they entail the same set of observational rank constraints, and the local violation of Condition 1 does not harm the correctness of other substructures. The output is also informative as $\mathcal{G}_2'$ is a compact representation of the rank-equivalence class that contains $\mathcal{G}_2$; one can easily infer many other members of the class from $\mathcal{G}_2'$.

(iii) The result is incorrect, but from the result we can infer that the conditions are violated. An example is given in Figure 20 (e) where for the v structure in $\mathcal{G}_3$ we have $|\{L_1, L_2\}| + |\{L_3\}| > |\{X_1\}| + |\{X_8\}|$ and thus Condition 2 is not satisfied. In this scenario, RLCD will output $\mathcal{G}_3'$, which is incorrect. However, we can easily infer that this result is abnormal, as the result is not consistent with the CI skeleton: in $\mathcal{G}_3'$, there are 3 subgroups that are not connected to each other, while the CI skeleton would inform us that all the variables are directly or indirectly connected.

We note that the above analysis holds in the large sample limit. In the finite sample cases, the result of statistical tests could be incorrect and thus self-contradictory.

### D.3 ILLUSTRATIVE OUTPUTS OF EACH METHOD.

Here we give illustrative outputs of each method to give an intuitive understanding of why the proposed RLCD performs the best. Here we use 10000 data points and only show the skeleton for simplicity.

The underlying graph $\mathcal{G}$ is shown in Figure 21 (a) and it has 16 observed variables and 4 latent variables. The output of the proposed RLCD, $\mathcal{G}_1$ is given in Figure 21 (b) and it is nearly the same as the input $\mathcal{G}$. As it is the finite sample case, there are still two edges missing due to the error of the statistic test. However, by increasing the sample size, the missing two edges can also be recovered.

The output of Hier. rank, $\mathcal{G}_2$ is given in Figure 21 (c). It introduces redundant latent variables compared to the ground truth, and many edges are incorrect. The underlying reason is that Hier. rank cannot handle the scenario where observed variables are directly adjacent and it does not allow edges from an observed variable to a latent variable.

The output of FCI, $\mathcal{G}_3$ is given in Figure 21 (d). It can neither identify the cardinality nor the location of latent variables. Furthermore, many edges between observed variables are missing or incorrect. This might be due to the fact that FCI relies on CI tests conditioned on multiple variables but tests of independence conditional on large numbers of variables have very low power Spirtes (2001).

The output of GIN, $\mathcal{G}_4$ is given in Figure 21 (e). The number of latent variables it discovered is correct, but they do not exactly correspond to the latent variables in the ground truth $\mathcal{G}$. Plus, many edges are incorrect, and the reason is similar to that for Hier. rank, i.e., GIN cannot handle direct edges between observed variables and edges from an observed variable to a latent variable

### D.4 DETAILED DISCUSSION ABOUT COMBINING RESULT OF PHASES 2 & 3 WITH PHASE 1.

(i) "Transfer the estimated DAG $\mathcal{G}''$ to Markov equivalence class" (the first part of Line 7 in Algorithm 1): Here, we transfer the output of Phases 2 & 3, i.e., $\mathcal{G}''$, to a Completed Partial Directed Acyclic Graph (CPDAG), which represents the corresponding Markov Equivalence Class. Algorithms for transferring a DAG to CPDAG have been well studied (Chickering, 2002b;a; 2013) with implementations available from e.g., causal-learn (Zheng et al., 2023) or causaldag (Chandler Squires, 2018) python package.

(ii) "Update $\mathcal{G}'$ by $\mathcal{G}''$" (the second part of Line 7 in Algorithm 1): Note that the input to Phases 2 & 3 is $\mathbf{X}_\mathcal{Q} \cup \mathbf{N}_\mathcal{Q}$, and let $\mathbf{L}_\mathcal{Q}$ be the newly discovered latent variables during Phases 2 & 3. By Theorem 9, latent variables must be in treks between variables in $\mathbf{X}_\mathcal{Q}$. Therefore, we just need to consider the edges among $\mathbf{X}_\mathcal{Q} \cup \mathbf{L}_\mathcal{Q}$. Specifically, we first delete all the edges among $\mathbf{X}_\mathcal{Q}$ in graph $\mathcal{G}'$, then add $\mathbf{L}_\mathcal{Q}$ to $\mathcal{G}'$, and finally add all the edges among $\mathbf{X}_\mathcal{Q} \cup \mathbf{L}_\mathcal{Q}$ from $\mathcal{G}''$ to $\mathcal{G}'$.

(iii) "Orient remaining causal directions that can be inferred from v structures" (Line 8 in Algorithm 1): Given $\mathcal{G}'$, a partially directed acyclic graph (PDAG), here we orient all remaining causal directions that can be determined. This can be easily achieved by following stage 2 of PC (Spirtes et al., 2000): we use the CI results detected in Phase 1 to find v-structures among observed variables and apply Meek's rule (Meek, 2013) to infer the remaining directions that can be decided. After that, we then transfer this PDAG to a CPDAG (well studied in Chickering (2002b;a; 2013) with implementations available) to get the corresponding Markov equivalence class.

