# OpenReview forum: "A Versatile Causal Discovery Framework to Allow Causally-Related Hidden Variables"
_ICLR.cc/2024/Conference — ICLR 2024 poster_

### Official Review · Reviewer_Rcya · 2023-10-26

**Soundness:** 3 good
**Presentation:** 4 excellent
**Contribution:** 4 excellent
**Rating:** 8
**Confidence:** 3

**Summary:**

This paper presents a new framework for causal discovery in the presence of unmeasured confounders and linear structural causal models (SCMs).

**Strengths:**

Overall, I like this paper, and I think the contribution made in this paper is quite huge.

1. The introduction is written in a straightforward and neat manner. The contribution is expressed clearly.
2. All the results seemed technically sound to me.
3. Every result is accompanied by examples, which greatly helps understanding the paper.
4. Experimental studies are conducted extensively, providing strong empirical benefits.

**Weaknesses:**

One minor weakness/limitation of the paper is that its method is confined to linear SCM. If the variables are mixtures of discrete and continuous random variables, or if they follow a nonlinear SCM, are there any opportunities for the proposed method to contribute?

**Questions:**

1. It’s a minor comment. I think the definition of $\Sigma_{\mathbf{A},\mathbf{B}}$ should be within Theorem 3.
2. If $\mathbf{A},\mathbf{B}$ are not t-separable, what does Theorem 3 imply?
3. I am curious about the benefits of the proposed algorithm in terms of running speed and time complexity compared to other existing algorithms, such as FCI or LiNGAM.

---

> ### Author Response · Authors · 2023-11-18
> **Author Response**
>
> We appreciate your time dedicated to reviewing our submission and your valuable feedback. Please see our responses to your questions point-by-point below.
>
> Q1: Are there any opportunities for the proposed method to be extended to nonlinear cases?
>
> A1: Thank you for your insightful question. Intuitively speaking, the rank of the covariance matrix serves as a certain kind of "information bottleneck", indicating the presence and dimensionality of mediators or common causes between two sets of variables. From this perspective, the proposed method has the potential to be extended to non-linear scenarios. Such an extension would require a non-linear version of 'rank' information, possibly through the utilization of kernel representations, and we will leave it for future exploration.
>
> Q2: If $\mathbf{A},\mathbf{B}$  are not t-separable, what does Theorem 3 imply?
>
> A2: By the definition of t-separation, a pair of sets of variables $\mathbf{A},\mathbf{B}$ are always t-separable: at least $\mathbf{A},\mathbf{B}$ can be t-separated by $(\mathbf{A},\emptyset)$ or $(\emptyset,\mathbf{B})$. Combining this with Theorem 3, we will arrive at $\text{rank}(\Sigma_{\mathbf{A},\mathbf{B}})\leq \min(|\mathbf{A}|,|\mathbf{B}|)$ which is consistent with the rank of a general matrix. We thank the reviewer for pointing out the confusion and will add this discussion to the revision.
>
> Q3: Definition of $\Sigma_{\mathbf{A},\mathbf{B}}$ within Theorem 3 and runtime analysis compared to methods such as FCI and LiNGAM.
>
> A3: We appreciate the reviewer's suggestion on writing and have made the corresponding revisions. Regarding the runtime analysis, it can be found in Appendix B.16. Specifically, the proposed method requires approximately 3 hours to complete all experiments (across three random seeds and three different sample sizes). FCI takes approximately 10 minutes, whereas RCD and a LiNGAM-based method each take around two days. In comparison to FCI, as we explore a much larger hypothesis space aiming to uncover the entire underlying causal structure (while FCI only indicates the existence of latent variables), RLCD is slower than FCI, but considerably faster than LiNGAM-based methods.
>
> We hope that your concerns/questions are well addressed and we would appreciate it if you had further feedback.

---

> > ### Comment · Reviewer_Rcya · 2023-12-01
> > **Response**
> >
> > I thoroughly read the comment and I am satisfied. Let me preserve the score as 8.

---

### Official Review · Reviewer_aySa · 2023-10-28

**Soundness:** 3 good
**Presentation:** 3 good
**Contribution:** 3 good
**Rating:** 6
**Confidence:** 4

**Summary:**

The authors propose a method for causal discovery in the presence of latent variables (`RLCD`), which makes use of observational rank information under the assumption of an underlying linear latent causal model, in which measured variables can be adjacent. The authors show how the rank information can be used to locate the presence of hidden variables and how their approach can reliably discover the causal structure over both measured and latent variables, asymptotically up to their Markov equivalence class.

**Strengths:**

The work is rigorously described for the most part and quite technical, yet the authors have managed to present intricate information in a reasonably clear manner. The approach seems to be an original extension of previous work on hierarchical structure and has potentially significant application, since it allows the user to uncover part of the latent causal structure under relatively mild assumptions. The differences between the proposed method and related work, including the PC algorithm and methods for identifying latent hierarchical structures, is clearly explained.

**Weaknesses:**

The paper is rigorously written for the most part, but I think there some parts are missing important details. For example, the steps in Algorithm 1 after Phases 1-3 are not described in detail, specifically how the cluster information is aggregated after the first three phases. It is also not very clear how the additional information is added into the CI skeleton.

I also think a bigger focus on the evaluation in the main paper would be warranted, since the experimental section seems a bit sparse. It is not quite clear to me why and where RCLD performs so much better than the competitors, just by looking at the F1 score. It would have been helpful to see the output of these different algorithms on a running example.

*Miscellaneous comments:*
- some references are duplicated (Judea Pearl - *Probabilistic reasoning in intelligent systems*, Shohei Shimizu et al. - *A linear non-gaussian acyclic model for causal discovery*.
- page 1, first paragraph in introduction: "ICA-based techniques ... that further **leverage**"
- page 2: repetition in "our main contributions are mainly three-fold"
- page 3: instead of the unusual construction ", - we basically", I would employ a semicolon, or simply start a new sentence.
- page 3, Section 3.2: I would not start a sentence with "E.g", but instead say 'For instance,'
- page 5, Algorithm 1 is introduced too early in the paper. I would move it to Section 4.2, where it is first explained.
- page 5, before Theorem 7: "uesful"
- page 6, Figure 3 is introduced too early in the paper. I would move it to Section 4.2, where it is first used to explain Alg. 1
- page 6, Figure 3(c) caption: "**Take** variables from..."
- page 6, Condition 1: "triangle structure" should be defined
- page 7, Section 4.2, second paragraph: I believe "Condition (i)-(iv)" is supposed to be **conditions (i)-(iv)**.
- page 7, last paragraph before Section 4.4: "We further determine the **neighbour** set... "
- page 7, last paragraph: "we **increase** *k* by 1"
- page 8, Table 2 appears too early in the paper, since the experiment section is on the next page.
- page 9, Section 7: "causal discovery approach that allows causally-related variables" seems to be an unfinished thought

**Questions:**

1. I am not convinced about the claim on page 1 that, for algorithms like FCI, "the research relies on the assumption that all hidden variables are independent of each other". Could you perhaps point to where that assumption is made? As far as I know and have checked, any maximal ancestral graph (MAG) is learnable from conditional independence using FCI, and a MAG is also valid if obtained from marginalizing over dependent latent variables.
2. By "latent ones" do you mean latent variables? If so, I would say latent variables instead, because it is not immediately clear what "ones" refers to.
3. In Theorem 7, what is the distinction between $\mathbf{X}$ and $\mathbf{X}_\mathcal{G}$?
4. Why is it so important to have a unified causal discovery framework, in the sense that rank constraints are used for finding the CI skeleton in Phase 1 of the procedure? I imagine it is more important to find an accurate CI skeleton, so do rank constraints provide more accurate *d*-separation statements than conditional independence tests? Could it also be better to mix different types of tests?
5. I imagine FCI would perform quite poorly in terms of the skeleton F1 score for all variables (Table 2), since it does not explicitly identify any latent variables, but how do you explain a score of 0.00? Does that mean that FCI did not get any edge right at all, not even between observed variables, as Table 3 also suggests?
6. What happens after Phase 3 in Algorithm 1? How is the information from $\mathcal{G''}$ transferred to the Markov equivalence class? How are the rest of the orientations performed? I am also confused by the fact that $\mathcal{G'}$ is supposed to be the skeleton on the observed variables (output of Algorithm 2 from PC), yet toward the end it becomes the MEC over both observed and latent variables (output of Algorithm 1). What am I missing here?
7. The idea of learning part of the latent structure explicitly has important ramifications. Could the authors perhaps comment on what extra information can be ascertained relative to methods like FCI, for which the latent structure is implicit? Put a different way, does the difference lie solely in the fact that some latent variables can be identified? Will RLCD always provide more structural information than FCI or other CI-based causal discovery algorithms?

---

> ### Author Response · Authors · 2023-11-18
> **Author Response (Part 1)**
>
> We thank the reviewer for carefully reading the submission and providing valuable comments and suggestions. Please see our point-by-point responses below.
>
> Q1: Detailed description of steps after Phases 1-3.
>
> A1: We thank the reviewer for the valuable suggestion. Accordingly, we have added a more detailed description of these steps in Appendix D.5.
>
> Q2:  Illustrative outputs of different algorithms to see why the proposed method is better.
>
> A2: We appreciate your helpful suggestion. We have added illustrative examples to Appendix D.4 for comparing the outputs of various methods, together with a discussion about why the proposed method performs better. Please see the resubmission for details, and below is a summary:
> RLCD correctly recovers the structure asymptotically, while FCI cannot recover edges involving latent variables, and Hier. rank and GIN cannot recover edges between two observed variables and the edges from observed variables to latent variables.
>
> Q3: Does FCI rely on the independence of latent variables?
>
> A3: Thanks for your insightful comment.  Indeed, the correctness of the FCI's result, which does not indicate whether potential latent confounders are causally related, does not rely on that assumption.  (With that sentence, we intended to say that the result by FCI is not informative about the potential causal relations among latent confounders.)  We thank the reviewer for your thoughtful comment and have revised the first paragraph accordingly.
>
> Q4: F1 score and  FCI's performance.
>
> A4: An F1 score of zero indicates that none of the edges are correctly identified. While, in theory, FCI can identify edges between observed variables up to a certain type of equivalence class, our experimental results with a finite sample size reveal that it does not perform well, as you noticed. There might be two main reasons: (i) the graphs we tested have relatively few direct edges between observed variables, and (ii) FCI relies on conditional independence (CI) tests that require conditioning on multiple variables. However, statistical tests for independence conditioned on a large number of variables have been reported to have low power [1], especially with relatively small sample sizes.
>
> [1] Spirtes, Peter. "An anytime algorithm for causal inference." International Workshop on Artificial Intelligence and Statistics. PMLR, 2001.
>
>
> Q5: In Theorem 7, what is the distinction between $\mathbf{X}$ and $\mathbf{X}_{\mathcal{G}}$?
>
> A5: As in Definition 1, $\mathbf{X_{\mathcal{G}}}$ refers to the set of all observed variables in the underlying graph $\mathcal{G}$, while $\mathbf{X}$ is a set of observed variables such that $\mathbf{X}\subseteq \mathbf{X_{\mathcal{G}}}$.
>
> Q6: By "latent ones" do you mean latent variables?
>
> A6: Yes. We thank the reviewer for pointing out this potential confusion and we have revised accordingly.
>
> Q7: Why is it so important to have a unified causal discovery framework, in the sense that rank constraints are used for finding the CI skeleton?  Do rank constraints provide more accurate d-separation statements than CI tests? Could it also be better to mix different types of tests?
>
> A7: We appreciate your insightful question. Theoretically, using rank constraints in specific ways following Lemma 10 and using CI tests would arrive at the same CI skeleton (for linear causal models). Also, empirically, we found that using CI tests and rank tests for Phase 1 would arrive at almost the same results (in terms of precision and recall of edges). We further looked into the type-2 errors of these two tests and found that the type-2 errors are also almost the same. For instance, in 1000 random trials each time using 1000 random samples, where H0 assumes independence and it is not true, when alpha=0.005 both the rank test and CI test have 70 type-2 errors. Given the substantial similarity in outcomes, we opted for the use of rank constraints to enhance the overall unity and coherence of the framework.

---

> ### Author Response · Authors · 2023-11-18
> **Author Response (Part 2)**
>
> Q8: What extra information can be ascertained relative to methods like FCI? Does the difference lie solely in that some latent variables can be identified? Will RLCD always provide more structural information than CI-based methods?
>
> A8: We thank the reviewer for the insightful question. It does not seem straightforward for extra information to be ascertained relative to methods like FCI, because CI relations among the observed variables are just a particular feature of the join distribution. On the other hand, rank constraints, if used in the way advocated in the paper, naturally capture more information about the distribution relevant to the underlying causal structure. Specifically, based on Lemma 10, we can see that all d-separations can be captured by rank constraints as special cases, that is, rank constraints are strictly more informative than CI information regarding the underlying structure.
>
> As for the comparison between algorithms, CI-based methods such as FCI can only infer edges between observed variables up to a partial ancestral graph, which may contain many uncertainties.  In contrast, the proposed RLCD offers more comprehensive information of the underlying structure: RLCD can identify all latent variables given the assumed conditions; at the same time, it can discover all causal edges among all observed or latent variables, without introducing extra edges (please note that some of the edges with double circles by FCI may not be causal edges, but just represent the existence of possible latent confounders).
>
> Q9: Miscellaneous comments (about the format, location of figures, etc.)
>
> A9: We are grateful for the helpful suggestions and have already revised the paper accordingly (we changed the location of the figures, corrected the typos, removed duplicated references, etc.).
>
>  We genuinely appreciate the reviewer's effort and hope that your concerns/questions are well addressed.

---

> > ### Comment · Reviewer_aySa · 2023-11-21
> > **Acknowledgment**
> >
> > Thank you to the authors for the detailed rebuttal and for addressing my concerns. I think the paper will significant benefit from the proposed additions.

---

> ### Author Response · Authors · 2023-11-22
> **Thanks and please kindly note that changes were made in the uploaded resubmission**
>
> Thank you for your positive feedback and we are happy that your concerns have been addressed.  Please kindly note that we have made those changes and uploaded the revised submission. It would be highly appreciated if you could have a look and see whether to update your recommendation at your earliest convenience.

---

### Official Review · Reviewer_G6SE · 2023-10-31

**Soundness:** 4 excellent
**Presentation:** 4 excellent
**Contribution:** 4 excellent
**Rating:** 8
**Confidence:** 4

**Summary:**

The paper presents a novel rank-based latent causal discovery algorithm to identify equivalence classes of directed acyclic graphs which can have both measured and latent variables. The causal relationships among the variables can be quite general compared to restricted patterns in the literature. They prove the discovery algorithm is asymptotically correct and degenerates to existing algorithms when certain aspects of the algorithm are simplified. Simulations and real data examples demonstrate the utility of their method.

**Strengths:**

1. The paper addresses a very important problem in causal discovery, i.e., identify graphs with latent variables
2. The graph pattern considered in the paper is quite general compared to existing methods.
3. The learning algorithm has theoretical guarantee in the large sample size limit.
4. Real-world example is quite convincing.
5. The paper is very well written.

**Weaknesses:**

No major weakness is found. Just a few minor ones; see the questions.

**Questions:**

1. Can the author explain the minimal-graph operator and the skeleton operator in addition to their definitions? Perhaps giving some examples will be helpful for readers to understand what equivalent graphs they entail.
2. Corollary 1 says it degenerates to PC when there is no latent variable. I suppose that is under the assumption that the causal model is linear as assumed throughout in this paper. In other words, if the truth is not linear, the PC is asymptotically correct but the proposed algorithm may not -- is it right?

---

> ### Author Response · Authors · 2023-11-18
> **Author Response**
>
> We are grateful to the reviewer for the insightful comments and valuable feedback. Please see our point-by-point responses to your comments and questions below.
>
> Q1: Explanations and examples of the minimal-graph operator and the skeleton operator will be helpful.
>
> A1: Thanks for the suggestion. As stated in the first paragraph of Section 5, an illustrative example with related explanations can be found in Appendix B.12. In light of your suggestions, we have added a brief description of the two operators in the first paragraph of Section 5 of the updated paper as well.
>
> Q2: Corollary 1 says it degenerates to PC when there is no latent variable. I suppose that is under the assumption that the causal model is linear as assumed throughout this paper.
>
> A2: Yes, you are totally correct, and we have made this statement explicit in the updated paper.
>
> Let us add that while we assume linear causal models throughout this paper, the framework has the potential to handle nonlinear relations. Intuitively, the rank of the covariance matrix serves as a certain kind of "information bottleneck", indicating the presence and dimensionality of mediators or common causes between two sets of variables. From this perspective, the proposed method has the potential to be extended to non-linear scenarios. This extension might involve incorporating a non-linear version of 'rank' information, possibly through the utilization of kernel representations. We thank the reviewer for the insightful comment and we will leave it for future exploration.
>
> We genuinely appreciate the reviewer's effort and hope that your concerns/questions are addressed.

---

### Official Review · Reviewer_DkWP · 2023-11-01

**Soundness:** 4 excellent
**Presentation:** 4 excellent
**Contribution:** 3 good
**Rating:** 8
**Confidence:** 4

**Summary:**

This work expands on the use of the rank of the covariance matrix to identify causal structures with latent variables in linear models. The authors give sufficient and necessary conditions under which latent variables of the causal graph can be identified using the rank. This insight is then used to devise an algorithm that can identify linear latent causal graphs upto an indeterminacy.

**Strengths:**

- Very well written and a pleasure to read. The paper is well structured and the exposition is clear.
- Claims are well justified.

**Weaknesses:**

- SID and SHD are more natural metrics for graphical evaluations
- Assumptions are not justified. (this might be for the field in general)
	- Assumptions are made on unmeasured variables and graphical structure that are hard to verify. Justification of why these are relatively weak (compared to previous work) would make this more useful.
- Relation to previous work does not contain enough information. It makes it hard to judge the exact contribution of this work. Some detail is given in the introduction.
	- Related work has been moved to the Appendix but there is not enough information about what the differences to similar works are. It would be useful if the most related works (e.g. Huang et al. 2022) are described in more detail.
	- For example, Hier. rank, that uses rank to discover hierarchical structures, is not described in sufficient detail. This leaves the question in the readers mind: what specifically allows for the identification of children of latents and mediatior latents etc as opposed to this work.

**Questions:**

- The assumptions in condition 1 are on unmeasured latent variables, how would you verify this before carrying out the graph search procedure?
- Similarly, how can you verify assumptions in condition 2? This seems particularly strong to me,
- Is Lemma 10 a contribution or has this been stated by previous work? If so, a reference is missing.
- Similar to the above, is Theorem 4 a contribution?

---

> ### Author Response · Authors · 2023-11-18
> **Author Response**
>
> We thank the reviewer for the time dedicated to reviewing the paper and the insightful comments. Please see our point-by-point responses below.
>
> Q1: Using metrics such as SID and SHD.
>
> A1: We thank the reviewer for the valuable comment. We have added new experimental results using SHD  in Appendix D.1, and we find that the proposed RLCD continues to outperform all baselines across all tested settings. For example, given 10K samples generated from latent measurement models, the average SHD of RLCD is 2.9, while the runner-up is 7.4. As for SID we are still running experiments and will include the results once they are available.
>
> Q2:  Can graphical assumptions be verified before the procedure? (might be for the field in general)
>
> A2: We appreciate your insightful comment. In practical applications, certain strategies can be employed to gain a preliminary understanding of whether the assumptions hold. One effective approach is to utilize domain knowledge of the data. For instance, if we know in advance that we have enough measurements of a latent variable that serves as pure children, then the atomic cover condition is satisfied. Nevertheless, we agree that a rigorous verification of graphical assumptions before the procedure serves as a  general challenge to the field and will be a focus for future research. At the same time, as illustrated below,
> in some cases one can verify the conditions after the procedure.
>
> Instead, traditional wisdom often focuses on another critical question: what implications arise if conditions are violated? We thank the reviewer for the valuable question and have provided a detailed discussion in Appendix D.2 with illustrative examples, which is summarized as follows.
> If a latent variable does not have enough pure children and neighbors, then Condition 1 is not satisfied. In this case, the result is still consistent with ground truth in the sense that they are rank-equivalent to each other, and at times, it can still provide valuable insights into the underlying structure. Moreover, if Condition 2 is not satisfied, then we will arrive at a self-contradictory situation, and thus we can infer that the condition is violated in hindsight.
>
> Q3: How is the condition weaker than previous method? More detailed discussion about the difference with Huang et al. 2022. For example, what specifically allows for the identification of children of latents and mediatior latents etc as opposed to this work.
>
> A3: A detailed discussion about how is our condition weaker than previous methods can be found in Appendix B.1 with illustrations in Figure 11.
>
> Compared to the most related work Hier. Rank (Huang et al. 2022), our graphical conditions are not only strictly weaker but also significantly weaker. Specifically, our conditions are strictly weaker in the sense that Hier. Rank can be taken as a restrictive, special case of the proposed method by prohibiting direct edges between observed variables, which is formally stated by our Corollary 1, and prohibiting direct causal influences from measured variables to latent ones. This is illustrated in Figure 11 (b) compared to Figure 11 (d).
>
> In practice, some measured variables may directly influence each other and they (e.g., environmental factors) may cause latent causal variables (which can cause other latent or measured variables). So our conditions are much weaker and hopefully open the gate to a much wider range of applications.
>
> Intuitively, our ability to identify latent structures that Hier. Rank cannot lies in the more flexible and comprehensive use of rank constraints. Specifically, Hier. Rank utilizes only a portion of the rank information, denoted as $\text{rank}(\Sigma_{\mathbf{A},\mathbf{B}})$, where $\mathbf{A}\cap\mathbf{B}=\emptyset.$ In contrast, our proposed method utilizes $\text{rank}(\Sigma_{\mathbf{A},\mathbf{B}})$ with arbitrary $\mathbf{A}$ and $\mathbf{B}$, which may share some variables in common, enabling the inference of more types of t-separations. This additional graphical information empowers us, e.g., to leverage Lemma 10 to identify edges between observed variables and Thm 8 to identify partially hidden and partially observed atomic covers.
>
> We thank the reviewer for the insightful question and have added the above discussion to the paper, at the end of Related Work.
>
> Q4: Are Lemma 10 and Theorem 4 novel contributions?
>
> A4: Yes, to be best of our knowledge, both of them are novel. Specifically, Theorem 4 provides a sufficient condition to determine nonadjacency, which generalizes PC’s condition to scenarios where latent variables may exist and Lemma 10 captures how partial correlation tests (as specific conditional independence tests) can be taken as a special case of rank constraints. As illustrated in the paper, they enable the proposed procedure to discover the underlying causal graph under much weaker assumptions.
>
> We genuinely appreciate the reviewer's effort and hope that your concerns/questions are addressed.

---

> > ### Comment · Reviewer_DkWP · 2023-11-22
> > **Response**
> >
> > Thanks for you response to the questions and the additional experiments. I will keep my score the same reflecting my view that this will make a good addition.

---

### Author Response · Authors · 2023-11-18
**Summary of the changes in the revised version**

Dear Reviewers,

We thank all the reviewers for your insightful review and valuable feedback.

According to your suggestions, we have revised our manuscript, with the changes being highlighted. The key updates can be summarized as follows: (1) We have included empirical results using SHD in Appendix D.1. (2) We have added a discussion to Appendix D.2, with illustrative examples, about the output of the algorithm when conditions are not satisfied. (3) We have included the description of the triangle structure in Appendix D.3. (4) We have added illustrative examples to compare the outputs of different methods in Appendix D.4. (5) We have added a detailed description of steps after Phase 1-3 in Appendix D.5. (6) We have incorporated all reviewers' valuable suggestions in writing.

In addition, please refer to our detailed responses to each reviewer below.

With best regards,\
Authors of #4264

---

### Public Comment · ~Tao_Feng2 · 2024-05-15
**please release code**

Please release the code and dataset.

---

> ### Public Comment · ~Xinshuai_Dong1 · 2025-02-03
>
> Thank you very much for your interest! We have recently optimized our code with multi-processing, making it ten times faster. The code and data will soon be available at https://github.com/dongxinshuai/scm-identify. We appreciate your patience, hope you find it useful and look forward to your feedback!

---

### Meta-Review · Area_Chair_B2T4 · 2023-12-05

**Metareview:**

The paper proposes an extension of some of the existing linear hierarchical latent variable models using the rank connection established by Sullivan et al. The reviewers are enthusiastic about the clarity, theoretical findings of the paper and experimental evaluations.

Based on my own reading, there was no mention of Gaussianity assumption which is necessary for the iff results of Sullivant et al that this paper builds on. This is a crucial assumption that needs to be added to the camera ready. I trust that authors will add this important limitation of their work in the camera-ready. I trust that this is not going to affect the rest of the paper, which is why I will recommend acceptance.

**Justification For Why Not Higher Score:**

After my own reading of the paper, I am slightly concerned that the authors leverage results from Sullivan et al. which relies on Gaussianity assumption, but do not make this assumption themselves. The main technical heavy lifting was done by Sullivan et al (and some results from Huang et al) so this seems quite an important point that was overlooked by all reviewers.

I do not believe the rank of covariance matrix is necessary and sufficient in general to inform conditional independence, thus Theorem 3 taken from Sullivan et al. should really make the Gaussianity assumption. The authors do not mention this and may have missed this important limitation of their work.

**Justification For Why Not Lower Score:**

The reviewers praised the importance of the problem, clarity of exposition, and that the work clearly extends the setup of linear latent hierarchical models.

---

### Decision · Program_Chairs · 2024-01-16

Accept (poster)